# Identification of Host Factors Associated with the Development of Equine Herpesvirus Myeloencephalopathy by Transcriptomic Analysis of Peripheral Blood Mononuclear Cells from Horses

**DOI:** 10.3390/v13030356

**Published:** 2021-02-24

**Authors:** Lila M. Zarski, Kim S. Giessler, Sarah I. Jacob, Patty Sue D. Weber, Allison G. McCauley, Yao Lee, Gisela Soboll Hussey

**Affiliations:** 1Department of Pathobiology and Diagnostic Investigation, Michigan State University, East Lansing, MI 48824, USA; zarskili@msu.edu (L.M.Z.); giessle1@msu.edu (K.S.G.); jacobs67@msu.edu (S.I.J.); allisongdavis@ymail.com (A.G.M.); leeyao@msu.edu (Y.L.); 2Department of Large Animal Clinical Sciences, Michigan State University, East Lansing, MI 48824, USA; weberp@msu.edu

**Keywords:** EHV-1, EHM, pathogenesis, herpesvirus, horse, PBMC, transcriptomics, RNA sequencing, microRNA, gene expression

## Abstract

Equine herpesvirus-1 is the cause of respiratory disease, abortion, and equine herpesvirus myeloencephalopathy (EHM) in horses worldwide. EHM affects as many as 14% of infected horses and a cell-associated viremia is thought to be central for EHM pathogenesis. While EHM is infrequent in younger horses, up to 70% of aged horses develop EHM. The aging immune system likely contributes to EHM pathogenesis; however, little is known about the host factors associated with clinical EHM. Here, we used the “old mare model” to induce EHM following EHV-1 infection. Peripheral blood mononuclear cells (PBMCs) of horses prior to infection and during viremia were collected and RNA sequencing with differential gene expression was used to compare the transcriptome of horses that did (EHM group) and did not (non-EHM group) develop clinical EHM. Interestingly, horses exhibiting EHM did not show respiratory disease, while non-EHM horses showed significant respiratory disease starting on day 2 post infection. Multiple immune pathways differed in EHM horses in response to EHV-1. These included an upregulation of IL-6 gene expression, a dysregulation of T-cell activation through AP-1 and responses skewed towards a T-helper 2 phenotype. Further, a dysregulation of coagulation and an upregulation of elements in the progesterone response were observed in EHM horses.

## 1. Introduction

Equine herpesvirus-1 (EHV-1) affects horses worldwide. It is the cause of upper respiratory disease, late term abortion or the crippling neurologic disease equine herpesvirus myeloencephalopathy (EHM). While respiratory disease can contribute to temporary loss of training, severe disease manifestations such as EHM can have devastating effects on animal welfare and an economic impact [1]. In a recent U.S. outbreak, as many as 14% of infected animals died or were euthanized due to EHM and to date, no vaccine is effective in preventing EHM [2,3]. This means strict biosecurity and quarantines during an outbreak remain the most effective strategy to control losses due to EHM. In order to limit these devastating consequences, the current recommendation is that state animal health officials issue quarantines of premises with EHV-1 confirmed cases for 21 days from the onset of the last case of EHM [4].

Following primary exposure, EHV-1 establishes a latent lifelong infection in the sensory ganglia or peripheral blood mononuclear cells (PBMCs) [5,6,7,8]. Despite this early natural infection and/or vaccination, horses remain susceptible particularly to secondary disease following a repeat exposure or reactivation of latent virus. During an acute infection, EHV-1 replicates in the nasal epithelium and causes upper respiratory disease in younger horses [9]. In the period of 4−14 days following exposure, EHV-1 is detectable in the PBMCs and transported throughout the body during a period of cell-associated viremia [10]. Viremia is a prerequisite event in the pathogenesis of EHM and facilitates the transfer of the virus to the vascular endothelial cells of the central nervous system [11,12]. It has been demonstrated that the duration and magnitude of the viremia contribute to the likelihood of EHM development, which suggests that increasing the exposure of the vascular endothelium to infected PBMCs increases the likelihood of CNS damage [13,14,15,16]. In further support of this, studies have shown that a single nucleotide polymorphism in the DNA polymerase gene of the virus, which encodes an aspartic acid residue at position 752 (genotype D752), has been associated with more neurologic disease during outbreaks than that which encodes an asparagine at this position (genotype N752) [1,17]. Challenge infection experiments in horses have confirmed that infection with the neuropathogenic D752 strain is associated with a higher magnitude and longer duration of EHV-1 viremia when compared to infection with the lower neuropathogenic N752 strain, and this corresponded to the increased likelihood of EHM [13,14,15,16,18].

Clinical EHM is typically observed towards the end of the viremic phase, suggesting that endothelial infection most likely occurs during peak levels of viremia [19]. Following infection of the vascular endothelium it is thought that a vasculitis, thrombosis, and neu-ronal damage develop and lead to clinical neurologic disease. Because of this, dysregulation of the blood−brain barrier (BBB) integrity is considered to be a critical event for clinical neurological disease during EHM. The BBB is comprised of the vascular endothelial cells of the CNS microvasculature and are connected with tight junctions and adherens junctions, forming a protective barrier for neurons [20,21]. The intact BBB protects against neuronal damage resulting from leukocytic infiltration [20]. Due to similarities in the damaging events at the BBB, the pathogenesis of EHM can be compared to that of human ischemic stroke. In human and mouse models of stroke, BBB integrity and permeability can be damaged as a result of an inciting event, such as hypoxia (via thrombosis) [20,22,23,24]. In horses, the coagulation cascade is known to be induced during EHV-1 viremia, and it is thought that this is a contributor to hypoxic BBB damage and neuronal damage resulting from microthrombosis [25,26,27,28]. In addition to ischemic damage via thrombosis, inflammation is another key factor in the development of BBB damage and EHM.

However, while viremia is a key feature of EHV-1 infection, and a large percentage of infected horses become viremic, only a small percentage of infected horses go on to develop clinical EHM. While some viral factors have been identified to increase the neuropathogenic potential of EHV-1 infection, it is clear that there are a number of host factors involved in the response to EHV-1 that correlate with the development of EHM [14]. While significant respiratory disease is more common in younger horses, it has been shown that older animals are more likely to develop EHM [13,19,29,30]. In addition, there is some evidence, that in mares over 20 years of age the incidence of clinical EHM is up to 70% [13,31]. The propensity of older animals to get EHM may be related to immunosenescence, which is a commonly observed feature of aging. In horses as well as humans this immunosenescence involves reduced levels of naïve lymphocytes, an increase in the memory lymphocytes, and the reduction of memory T-cell activation and proliferation following stimulation [32,33]. Additionally, aging is often associated with low levels of chronic inflammation, known as “inflammaging,” which is thought to contribute to the pathogenesis of other inflammatory diseases of older humans and horses [32,33].

In order to identify which host factors are associated with the development of EHM, we thought to utilize the “old mare model” to reliably induce EHM and then compare which host factors are affected in horses with clinical EHM compared to horses that are infected and viremic but do not exhibit clinical EHM. For this, we conducted a challenge infection with a neuropathogenic strain of EHV-1 in both young and old horses in order to ensure enough animals presented with both phenotypes (those that progressed to clin-ical EHM and those that did not). Based on the central role of viremia for systemic host immunity and EHM pathogenesis and in induction of BBB dysfunction, the goal of the study was to use deep RNA sequencing and a repeated measure design to identify the differences in the host and virus transcriptome in PBMCs of horses that did and did not develop clinical EHM following an EHV-1 challenge infection.

## 2. Materials and Methods 

### 2.1. Animals

Fourteen horses of mixed breeds (5 males, 9 females) were used in this study. The study consisted of two challenge infection experiments that occurred several months apart. One experiment used 2-year-old horses (n = 7; 5 males, 2 females) and the second experiment was performed in aged horses (18–21-year-old; n = 7; 7 females). For both experiments, animals were housed in a building with natural ventilation with multiple horses per pen and nose-to-nose contact between pens. Horses had access to grass hay and water ad libitum for the entirety of the study. All animal maintenance and procedures were performed in compliance with Michigan State University’s Institutional Animal Care and Use Committee, under protocol “PROTO201800015”.

### 2.2. Virus

The neuropathic strain EHV-1 Ab4 (GenBank Accession No. AY665713.1) was propagated in NBL-6 cells (ATCC CCL-57) with MEM-10 (Minimum Essential Medium Eagle [Sigma-Aldrich, St. Louis, MO, USA] supplemented with 100 IU/mL penicillin, 100 µg/mL streptomycin, 1% GlutaMAX (GIBCO, Life Technologies, Carlsbad, CA, USA), 1 mM sodium pyruvate, 1% nonessential amino acids [M7145, Sigma-Aldrich], and 10% fetal bovine serum). After incubation at 37 °C and 5% CO2 for 3–4 days, the cells were frozen and thawed, and cellular debris removed by centrifugation at 300× *g* for 10 min. The stock was stored at −80 °C. Prior to inoculation in horses, the stock was thawed and sonicated for three cycles of 30 s at 50% amplification.

### 2.3. Experimental Design and Sample Collection

Prior to inclusion in the study, the serum was screened for virus neutralizing antibodies for EHV-1. The 2-year-old horses had pre-challenge titers ≤1:8 for EHV-1 and the aged horses had pre-challenge titers of ≤1:32 for EHV-1. Horses were inoculated with in-tranasal instillation of 5 × 10^7^ pfu of EHV-1 Ab4. Physical examinations were conducted prior to challenge infection (CH) and daily following CH and included the evaluation of nasal discharge, ocular discharge, cough, and rectal body temperature as well as a neuro-logical exam using the simplified version of the Mayhew scale, as described by Allen [13]. Peripheral blood mononuclear cells (PBMCs) were isolated from horses prior to CH and daily for 10 days post CH. One hundred mL whole blood was collected in heparinized syringes via jugular venipuncture and immediately transported to the laboratory for processing. PBMCs were separated by density gradient centrifugation over Histopaque-1077 (Sigma-Aldrich, St. Louis, MO, USA) as previously described and cell pellets of 6 × 10^7^ were stored at −80 °C until RNA isolation [34]. An additional aliquot of 1 × 10^7^ PBMCs was used for quantification of viremia using qPCR as previously described [14]. The day post CH for peak viral load in PBMCs was determined for each horse and RNA for RNA sequencing analysis was isolated from PBMCs from each horse pre CH, as well as on the day of peak viremia. Statistical analysis was performed using a 2-way ANOVA with significance of *p* < 0.05.

### 2.4. RNA Isolation, Library Preparation, and Sequencing

Cell pellets were lysed and homogenized using TRIzol Reagent (Thermo Fisher Scientific Waltham, MA, USA) following the manufacturer’s instructions. The aqueous phase was then collected, washed with 100% ethanol and total RNA was isolated using the miRNeasy Mini Kit (Qiagen Inc., Hilden, Germany) according to the manufacturer’s instructions. To eliminate genomic DNA contamination, deoxyribonuclease treatment (Qiagen) was applied to each sample according to the manufacturer’s recommendation. The concentration of RNA was determined using fluorometric quantification with the Qubit 1.0 (Thermo Fisher). RNA quality was evaluated using the Agilent 2100 Bioanalyzer (Agilent Technologies, Inc., Santa Clara, CA, USA) with the RNA 6000 Pico Assay and samples with a RIN score >6.70 were submitted for sequencing. Library preparation and next generation sequencing were performed at Michigan State University’s Genomics Research and Technology Support Facility. Stranded mRNA cDNA library preparation was performed using the Illumina TruSeq stranded mRNA kit (Illumina Inc., San Diego, CA, USA) with IDT for Illumina Unique Dual Index adapters according to manufacturers’ recommendations. MicroRNA (miRNA) cDNA libraries were prepared using the Illumina TruSeq Small RNA Library Preparation Kit (Illumina Inc.) following manufacturer’s recommendations. Completed libraries were quality controlled and quantified using a combination of Qubit dsDNA HS (Thermo Fisher) and Agilent 4200 TapeStation HS DNA1000 (Agilent Technologies) assays. The stranded mRNA cDNA libraries were divided into 5 pools for multi-plexed sequencing; four of these pools contained 8 libraries and the fifth pool contained 7. Pools were quantified using the Kapa Biosystems Illumina Library Quantification qPCR kit (Roche, Basel, Switzerland). Each pool was load-ed onto one lane of an Illumina HiSeq 4000 flow cell (Illumina Inc.) and sequencing was performed in a 2 × 150 bp paired end using HiSeq 4000 SBS reagents. (Illumina Inc.) The miRNA cDNA libraries were divided into two pools and the pools quantified using the Kapa Biosystems Illumina Library Quantification qPCR kit (Roche). Each pool was loaded onto one lane of an Illumina HiSeq 4000 flow cell (Illumina Inc.) and sequencing was performed in a 1 × 50 bp single read format using HiSeq 4000 SBS reagents (Illumina Inc.). Base calling was done by Illumina Real Time Analysis v2.7.7 (Illumina Inc.) and output of RTA was demultiplexed and converted to FastQ format with Illumina Bcl2fastq v2.19.1 (Illumina Inc.). All raw sequencing reads are available in the NCBI sequence read archive (SRA, NCBI, Bethesda, MD, 20894 USA) under BioProject ascension number PRJNA705083.

### 2.5. Genome Guided mRNA Alignment

Read quality was assessed before and after quality and adaptor trimming using FastQC software (version 0.11.7, [35]). Illumina adapters were trimmed from files using Trimmomatic (version 0.38, [36]) with the additional options for quality trimming: LEADING:2, TRAILING:2, SLIDINGWINDOW:4:2, and MINLEN:25. Reads were mapped to the Equus caballus genome (assembly EquCab3.0, ENSEMBL release-95) using HISAT2 (version 2.1.0, [37]). The accepted hits.BAM files were sorted by name using SAMtools (version 1.5, [38]), and read counts generated using htseq-count ([39]) (built in with Python version 3.6.4) with the following options: —format = bam, —stranded = reverse, and —order = name.

### 2.6. Host and Viral miRNA Identification and Quantification

Read quality was assessed before and after quality and adaptor trimming using FastQC software v0.11.7 [35]. Raw reads were trimmed using cutadapt (version 1.16, [40]). Options included trimming the Illumina adaptor sequence (option -a TGGAATTCTCGGGTGCCAAGG), reads shorter than 15 bp dis-carded (option -m 15), and the 3′ end trimmed with a quality score cutoff of 20 (option -q 20). After trimming, miRDeep2 (version 2.0.0.8, [41]) was used to identify all miRNAs present in the samples, including putative novel miRNAs, as well as known miRNAs. For this, reads from all samples were first pooled. Next, a combined reference genome was created from combining the horse (EquCab3.0; down-loaded from ENSEMBL release-95) with the genome of the four most common equine herpesviruses (EHV-1 NCBI RefSeq NC_001491.2, EHV-2 NCBI RefSeq NC_001650.2, EHV-4 NCBI RefSeq NC_001844.1, and EHV-5 NCBI RefSeq NC_026421.1) and indexed using the bowtie-build function of Bowtie (version 1.2.2., [42]) The mapper.pl function of miRDeep2 was used on the pooled reads to the reference genome to create a .fasta file with processed reads and a .arf file with mapped reads. Next, miRDeep2.pl function was performed on these outputs using equine miRNAs as the main reference. Since the equine miRNA database is still very incomplete, mouse and human known miRNAs were used as related species lists. Known equine, mouse, and human miRNAs were downloaded from the miRbase database (release 22.1, [43]). Novel RNAs with a miRDeep score < 1 were removed for subsequent analysis.

In order to perform differential gene expression analysis, the miRNAs were quantified in each sample. For this the quantifier.pl tool from the miRDeep2 package was used. All novel and known mature and precursor miRNAs from the miRDeep2 step above were used as input reference sequences. Quantifier.pl was run with the processed reads .fasta file as input and the -k option was used to consider precursor-mature mappings that have different ids. Predicted target genes of all differentially expressed miRNAs were iden-tified using the miRDB target search tool [44].

### 2.7. Differential Gene Expression Analysis

An overview of the data analysis pipeline is described in Figure 1 (mRNA) and Figure 2 (miRNA). A repeated measure study design was used to assess differential gene expression in response to EHV-1 CH both between groups (EHM vs. non-EHM) and with-in groups (pre vs. post CH). The between group, “contrast” comparison was performed which compared the response (pre vs. post) in the EHM group to the response (pre vs. post) in the non-EHM group in order to identify differences between the EHM and non-EHM horses that contribute to or protect from EHM. To expand on this and identify additional potential contributing or protective mechanisms, we also performed the within group comparison. For this, we determined which genes were up- or downregulated (pre vs. post CH) uniquely in either non-EHM (protective from disease) or EHM horses (contributing to disease).

Differential expression analyses for both mRNAs and miRNAs were performed using the edgeR package (version 3.24.3, [45]), in R (version 3.5.3). We analyzed differences between groups (the “contrast” comparison of EHM vs. non-EHM) and within groups (prior to and after EHV-1 CH for each group) using a repeated measure type design as described in Section 3.5 of the EdgeR manual (last revised 2020, [46]). For this, the model matrix was designed as: ~group + group:horse + group:timepoint, where “group” indicated normal or EHM, “horse” were individual horses, and “timepoint” referred to pre- or post EHV-1 CH. Preprocessing of the data were performed as recommended in the edgeR user manual and included filtering to eliminate low expressed genes (filterByExpr function), TMM normalization (calcNormFactors function), and dispersion estimations (estimateDisp function). We then fit a genewise general linear model (GLM) using the glmFit function, and then performed likelihood ratio testing (LRT) for comparisons using the glmLRT function. Three LRT comparisons were made as explained above: genes that respond differently to the virus in EHM horses compared to how they respond in normal horses (contrast comparison between groups), genes responding to the virus in EHM horses (within group), and genes responding to the virus in non-EHM horses (within group). Benjamini−Hochberg adjustment was used on the *p* values and statistical significance was set at an FDR < 0.05 and log fold change >|1|. PANTHER and UniProtKB databases were used to identify families and functions of proteins associated with orthologous human genes [47,48].

### 2.8. Gene Ontology (GO) Enrichment Analysis

Enrichment analysis for gene ontology (GO) terms was performed on the gene lists derived from the differential expression analysis in order to meaningfully interpret and consolidate the up- or downregulated gene lists and identify relevant functions associ-ated with these genes. GO is a system of classification for genes based on their biological functions, and genes can be classified to any number of GO terms. Overrepresentation analysis identifies which of the GO terms are statistically more likely (or “over-represented”) based on the proportion of genes in the data set compared to the proportion of all genes in that species that are classified to a certain term.

Horse ENSEMBL IDs were converted to gene symbol based on the annotations in the EquCab3.0 GTF file (ENSEMBL release-95). GO enrichment analysis for biological processes was performed using the enrichGO function from the clusterProfiler package [49]. The database used was the human “org.Hs.eg.db”, and the background consisted of all genes present in our samples [50]. *p* values were adjusted using the Benjamini−Hochberg correction and statistical significance set at *p* < 0.05. After generating lists of enriched GO terms, redundant terms were removed using REVIGO [51]. Enriched GO terms, along with their associated adjusted *p* value were provided as input, and the allowed similarity was small (0.5). The default settings were used, which included selecting the whole UniProt database to determine GO term sizes and using the SimRel semantic similarity measure for the analysis.

### 2.9. In Silico Cell Sorting

Cell fractions were imputed using CIBERSORTx [52]. The included LM22 (22 immune cell types) was used as the signature matrix file. The mixture file included the CPM values for each gene from all of our samples, identified with gene symbol, and CPM values of redundant genes were merged together. B-mode batch correction was performed, and quantile normalization was disabled (recommend for RNA-seq data). Permutations for significance analysis was set at 100. The run was performed in relative mode (default). The data was determined to be non-normally distributed as determined by Shapiro−Wilk testing. Therefore, a Wilcoxon signed rank test was used to compare the paired data for each group.

### 2.10. Whole Blood Cytokine RT-qPCR

In order to validate changes in cytokine mRNA expression between groups, RNA was isolated from whole blood collected via jugular venipuncture into PAXgene RNA Blood Tubes (BD Biosciences, San Jose, CA, USA). Samples were collected prior to the EHV-1 CH and on day 7 post CH. RNA was isolated following the manufacturer’s instructions. RNA was reverse transcribed using the High-Capacity cDNA Reverse Transcription Kit with RNAse inhibitor (Applied Biosystems, Waltham, MA, USA). Real time PCR was performed using the SmartChip Real-Time PCR System (Takara Bio Inc., Kasatsu, Shiga, Japan) following the manufacturer’s recommendations. The reactions consisted of template cDNA, TaqMan Gene Expression Master Mix (Applied Biosystems) and TaqMan Gene Expression assays for equine target genes (Thermo Fisher, Waltham, MA, USA). Target genes included *CCL5*, *CXCL10*, *IRF7*, *IRF9*, *MMP9*, *THBS1*, *GUSB*, *ACTB*, and *YWHAZ*. The negative delta delta Cq (-ddCq) was calculated following the Livak and Schmittgen method [53]. For this, three housekeeping genes (*GUSB*, *ACTB*, and *YWHAZ*) were averaged to normalize the gene of interest and the average of pre CH values for each group were used as calibrators. Statistical differences between -ddCq values between non-EHM and EHM groups were determined for each gene of interest using a Wilcoxon rank sum test in R and significance set at *p* < 0.05.

## 3. Results

### 3.1. Clinical Disease and Viremia Differed between EHM and Non-EHM Horses in Response to EHV-1 Challenge Infection

The clinical data presented in this section were collected as part of a separate study by our laboratory, but are being summarized here to provide the clinical context to the transcriptomic analysis presented in the current paper [54]. All horses were free from clinical signs of respiratory disease, had normal body temperatures, and were negative for EHV-1 genome in nasal swab and PBMC samples when tested by real-time PCR prior to CH with EHV-1. All horses developed fevers and shed virus in nasal secretions following EHV-1 inoculation, indicating successful challenge infection. The EHM group consisted of all seven of the horses from the aged horse group and one horse from the young group; all horses that developed EHM were female. Neurological symptoms in the EHM group appeared as early as 6 days post CH and all seven old horses developed severe ataxia/paralysis and were humanely euthanized on day 9 or 10 post CH. The one horse from the young horse group developed moderate ataxia and recovered from neurological symptoms by the end of the study. The non-EHM group consisted of the remaining young horses and did not develop neurological clinical signs (1 female and 5 males).

The fever response to EHV-1 infection is often characterized by a primary fever (corresponding to peak nasal viral shedding) and a secondary fever (corresponding to peak viremia) [10,55]. Looking at the body temperatures, all non-EHM horses showed a pri-mary fever immediately following CH (days 1–4), while all but one EHM horse did not develop a primary fever. Additionally, non-EHM horses developed more severe clinical respiratory disease (nasal discharge and cough) compared to EHM horses, which showed only very mild symptoms. Finally, non-EHM horses showed significantly more nasal viral shedding when compared to EHM horses. In contrast, EHM horses developed significantly greater viremia levels when compared to non-EHM horses. To select post CH samples for RNA sequencing, the day of peak viremia was identified for each horse and occurred days 6–10 post CH (Figure 3A,B).

### 3.2. mRNA Sequencing Revealed Differential Host Gene Expression between EHM and Non-EHM Horses

After paired-end sequencing, there was an average of 43,730,408 reads per sample, and 85.2% of total reads uniquely mapped to the EquCab3.0 genome (Appendix A). Principal component analysis of the regularized log transformed read-count data showed a clustering of samples, with a few exceptions, based on timepoint (pre CH vs. post CH) and group (EHM vs. non-EHM) (Figure 4).

A repeated measure study design was used to assess differential gene expression in response to EHV-1 CH both between groups and within groups. The between group comparison (contrast comparison) identified the differentially expressed genes between the response to the virus in EHM vs. non-EHM horses. This was performed in order to identify the potential host factors or biological responses that contributed to EHM pathogenesis. We then supported these findings by identifying additional genes that were uniquely found to be differentially regulated in either EHM or non-EHM horses in response to infection, but not in both groups. For this, we identified the within group responses by determining the response to CH (pre vs. post CH) in the EHM and non-EHM groups and then selecting which of these genes were unique to each group (unique to EHM or unique to non-EHM). Significance was set at FDR < 0.05 and log fold change >|1|.

For the between group comparison (contrast), there were 181 DEGs (37 upregulated and 144 downregulated in EHM horses, compared to non-EHM horses) (Figure 5A, Appendix A).

For the within group response for non-EHM horses, there were a total of 113 DEGs (109 upregulated during infection and 4 downregulated during infection) (Figure 5B, Appendix A). When looking at the within group response for the EHM horses, there were a total of 490 DEGs (239 upregulated during infection and 251 downregulated during infection) (Figure 5C, Appendix A).

In order to identify genes that were uniquely upregulated and downregulated in either group during EHV-1 infection, we used a Venn diagram (Figure 6). There were 93 commonly upregulated and 1 commonly downregulated gene shared between EHM and non-EHM horses. There were 146 uniquely upregulated and 250 uniquely downregulated genes in EHM horses (potential risk factors) and 16 uniquely upregulated and 3 uniquely downregulated in non-EHM horses (potentially protective factors) (Figure 6A,B).

### 3.3. Gene Ontology Overrepresentation Analysis Identified Enriched Biological Processes in EHM Horses

A list of all enriched GO terms for the between group contrast comparison and also for the within group comparison unique to EHM horses can be seen in Appendix A. Due to the limited number of uniquely regulated genes in non-EHM horses, GO overrepresentation analysis did not result in any enriched terms. After removing redundant GO terms using REVIGO analysis, for the between group contrast comparison, there were five enriched biological processes (with eight associated genes) from upregulated genes in EHM horses compared to non-EHM horses (Figure 7A and Figure 8A, Appendix A), and 12 enriched biological processes from the downregulated genes (with 36 associated genes) (Appendix A, Figure 7B and Figure 8B).

When looking at the genes differentially regulated in response to CH uniquely in EHM horses, a total of 12 biological processes were enriched (with 39 associated genes) and 7 biological processes downregulated (with 41 associated genes) (Table 1).

For the between group contrast comparison, the gene associated with the most biological processes upregulated in EHM horses compared to non-EHM horses was IL6, which was involved in 4/5 enriched biological processes (Figure 9A). IL6 encodes the interleukin-6 (IL-6) protein, which is implicated in several diseases involving immune mediated damage of the vascular endothelium and serves as an important biomarker in human stroke [56,57]. Additional upregulated responses included positive regulation of T-helper type 2 immune response (*IL6*, *RSAD2*), positive regulation of cytokine production (*NOX1*, *RSAD2*, *IL1RL1*, *IL6*, *LPL*) and regulation of cellular pH (*NOX1*, *SLC4A9*, *SLC9B1*) and upregulation of the interferon stimulated gene (*RSAD2*; Figure 9B). As seen in the between group contrast comparison, the gene involved with the most enriched process uniquely upregulated within EHM horses was also IL6. Additionally, upregulated functions and the associated genes were largely related to the defense response to virus and the type 1 interferon response pathway (*CYBB*, *ACOD1*, *ZBP1*, *IRF9*, *TGM2*, *NR1H3*, *IL15*, *IRF7*, *CCL8*, *ISG20*, *NMI*, *CD180*, *ISG15*, *TIMP1*, *ADAR*, *AIM2*, *EIF2AK2*, *CXCL10*, *RTP4*, *IL6*, *TRIM56*, *LPL*, *TLR3*, *LAG3*, *SLPI*) (Figure 9A,C).

For the processes that were downregulated in the between group contrast comparison, JUN and FOS, the genes which encode the subunits for the AP-1 transcription factor (also important for T-cell activation) were downregulated (Figure 9D,E). FOS (en-coding c-fos protein) and JUN (encoding c-jun protein) are protooncogenes well known for their role in cell proliferation [58]. The downregulation of FOS and JUN corresponded to the many other genes (*KLF4, INSR, JUN, PDGFB, FGFR1, CCL5, EPHA4, NLRP12, DUSP1, ATF3, DUSP6*) we observed downregulated that are known to be related to either positive or negative regulation of the mitogen-activated protein kinase cascades (such as the extracellular signal related kinase cascade) which are important in T-cell activation and proliferation [59]. Furthermore, a key proinflammatory cytokine gene expressed by activated T-cells, IFNG, was downregulated in EHM horses (Figure 9F). Other functions downregulated included leukocyte migration (*IFNG*, *KLRK1*, *TREM1*, *PLCB1*, *NLRP12*, *CCL5*, *MMP9*, *CD244*, *PDGFB*, *SLC7A11*, *DUSP1*, *CX3CR1*, *DAPK2*, *TBX21*), regulation of apoptosis (*ACER2*, *IFNG*, *KLRK1*, *JUN*, *NLRP12*, *CCL5*, *PLK2*, *ATF3*, *FGFR1*, *MMP9*, *KLF4*, *SLC7A11*, *DUSP1*, *CX3CR1*, *DAPK2*, *GSN*, *DUSP6*), regulation of transcription (*JUN*, *ATF3*, *KLF4*, *EGR2*, *FOS*, *TBX21*), regulation of cell adhesion (*KLF4*, *IFNG*, *INSR*, *CX3CR1*, *CCL5*, *EPHA4*, *PLCB1*, *DUSP1*, *SLC7A11*, *ACER2*), vascular endothelial cell growth and proliferation (*PDGFB*, *FGFR1*, *SLC7A11*), oxidative stress (*MMP9*, *SLC7A11*), and NK cell activation (*KLRK1*, *CD244*, *TBX21*). Additional downregulated functions were identified when evaluating the within group comparison for EHM horses including those associated with fibrinogen complex formation (*THBS1*, *FN1*), and additional JUN/FOS genes (*JUND* and *FOSB*).

Due to the limited number of uniquely regulated genes in non-EHM horses which may predict protective functions, GO overrepresentation analysis did not result in any enriched terms. Instead, we investigated the individual functions of the differentially expressed genes (Appendix A). These genes had different functions and included: upregulated (*BFSP2*, *NFE2*, *HEY1*, *CILP*, *SCD*, *CISH*, *TCF7L1*, *DUSP6*, *FAM111B*, *FRMD4A*, *LZTS1*, *SHISA5*) and downregulated (*CGA*, *SYTL2*, *WWTR1*) genes. General functions of the upregulated genes include positive or negative regulation of signal transduction (*CISH*, *TCF7L1*, *DUSP6*), regulation of cell growth (*LZTS1*), scaffolding (*FRMD4A*, *CILP*), repression of transcription (*HEY1*), and a transcription factor subunit for NF-E2 (*NFE2*). Functions of the downregulated genes include hormone signaling (*CGA*), cytotoxic granule exocytosis in lymphocytes (*SYTL2*), and negative regulation of cell proliferation (*WWTR1*).

### 3.4. In Silico Cell Sorting Identified Differences in Cell Population Fractions between EHM and Non-EHM Horses in Response to EHV-1 Challenge

Results of the in silico cell sorting are shown in Table 2. The most abundant cell type identified in all groups was naïve B-cells with 23–37% of the cell population identifying with this fraction. In EHM horses, we observed a higher percent (~8%) of CD8+ T-cells pre-infection than in non-EHM horses (~2%). Additionally, in EHM horses, there was an increase in percentage of M1 and M2 macrophages, resting dendritic cells, and eosinophils following infection with EHV-1, but a decrease in percentage of CD8+ T-cells, regulator T-cells, resting NK cells, M0 macrophages, and activated mast cells. In the non-EHM horses, there was an increase in plasma cells and CD4+ activated memory T-cells and a decrease in naïve B-cells and follicular helper T-cells. In both groups, there was an increase in percentages of γδ T-cells and activated dendritic cells (Table 2).

### 3.5. Whole Blood Cytokine RT-qPCR Confirmed Differential Expression of Select Genes between EHM and Non-EHM Horses 

In order to validate the differential gene expression results of our RNA sequencing analysis, RT-qPCR cytokine gene expression for select genes was determined from whole blood RNA collected pre CH and day 7 post CH. In agreement with RNA sequencing differential gene expression results from PBMCs, RT-pPCR from whole blood showed *CCL5*, *MMP9*, and *THBS1* were significantly downregulated in the EHM group when compared to non-EHM horses (Figure 10A,G,I). In agreement with RNA sequencing data, *CXCL10* and *IRF9* expressions were also significantly upregulated in the EHM group when compared to non-EHM horses (Figure 10C,E). *IRF7* was upregulated in EHM horses when compared to non-EHM horses, though this was not significant (*p* = 0.14). For a convenient comparison, the normalized count data obtained from the RNA sequencing analysis from PBMCs for these genes have been expressed as delta-CPM (post CH CPM value—group average of pre CH CPM) and can be seen in Figure 10.

### 3.6. Viral mRNA Sequencing Identified EHV-1, EHV-2, and EHV-5 Transcripts in PBMCs Prior to and after EHV-1 Challenge Infection

Normalized read counts expressed as transcripts per million (TPM) for EHV-1, EHV-2, and EHV-5 are found in Appendix A. No reads mapped to the EHV-4 genome. There was a low level of transcription of a single EHV-1 gene in two of the EHM horses prior to the EHV-1 CH (Appendix A). As expected, during EHV-1 viremia post CH EHV-1 transcripts were present in all samples. The most abundant EHV-1 genes expressed in EHM horses during peak viremia were *ORF34*, *ORF25*, *ORF18*, and *ORF75* (Figure 11). The products of these genes include a protein involved in the early step of virus egress (*ORF34*) and a capsid protein (*ORF25*), a DNA polymerase processivity factor (*ORF18*) and a membrane protein presumed to be involved in the virulence of certain EHV-1 strains (*ORF75*) [60,61]. The most abundant EHV-1 genes induced in non-EHM horses during viremia were *ORF34*, *ORF18*, *ORF51*, and *ORF42* (Figure 11). *ORF51* encodes the pUL11 protein which appears to have differing roles in viral replication depending on the strain; however, it has been shown to be essential for replication of strain Ab4 in cell culture [62,63]. *ORF42* encodes a capsid protein [60]. Additionally, transcription of the equine gammaherpesvirus (EHV-2 and EHV-5) genes were present in PBMCs prior to and post EHV-1 CH in both non-EHM and EHM groups (Appendix A). EHV-2 transcripts were identified in 1/8 and 2/8 EHM horses pre and post EHV-1 CH, respectively, and 3/6 and 4/6 non-EHM horses pre and post EHV-1 CH, respectively. EHV-5 transcripts were identified in 2/8 and 3/8 EHM horses pre and post EHV-1 CH, respectively, and 4/6 and 3/6 non-EHM horses pre and post EHV-1 CH, respectively (Appendix A).

### 3.7. Host, EHV-2, and EHV-5 miRNAs Were Identified in PBMCs of Horses

MiRDeep2, a software tool for miRNA mapping and identification, identified 285 known mature equine miRNAs amongst the pooled samples. Furthermore, we identified 962 total novel miRNAs with a miRDeep2 score > 1 (File S4). Of these novel miRNAs, 860 mapped to the equine genome, 52 mapped to the EHV-2 genome, and 50 mapped to the EHV-5 genome. For EHV-2, the miRNAs clustered around three general regions on the genome: 38–44 kb and 176–182 kb on the plus strand, and 125–127 kb on the minus strand. For EHV-5, the miRNAs clustered around two general regions: 36–43 kb on the plus strand and 125–127 kb on the minus strand (File S4). Interestingly, no miRNAs were identified that mapped to either of the equine alpha herpesviruses, EHV-1 or EHV-4 (File S4).

### 3.8. Host miRNAs Were Differentially Expressed between EHM and Non-EHM Horses in Response to EHV-1 Challenge Infection

Novel miRNAs with a miRDeep2 cutoff > 1 were added to the list of known miRNAs for quantification in each sample and quantification was performed to identify differentially expressed miRNAs in horses pre and post EHV-1 infection. There was an average of 14,122,680 reads per sample with an average mapping of 59.7% (Appendix A). Principal component analysis plot analysis of these counts indicated that samples clustered based on group (EHM vs. non-EHM), rather than within horses as a result of infection (Figure 12).

Thus, it was not surprising to have limited numbers of miRNAs differentially ex-pressed as a result of EHV-1 infection. For the contrast comparison looking at the differences in response to infection between EHM and non-EHM groups, two miRNAs were upregulated and seven miRNAs were downregulated in EHM horses compared to non-EHM horses (Table 3). For the within group comparisons, there were no miRNAs differentially expressed in non-EHM horses pre vs. post EHV-1 CH. In contrast, in the within group comparison for EHM horses, there were five miRNAs uniquely upregulated and five miRNAs uniquely downregulated in response to EHV-1 CH (Table 3).

All of the upregulated genes for both the contrast and within group comparisons identified as novel miRNAs. These included one with the murine ortholog *mmu-miR-7059-5p* which has been shown to be involved with the downregulation of immunoregulatory genes and pathways [64]. Another miRNA upregulated in EHM horses included one with the human ortholog *hsa-miR-7108-3p*, which has been shown to be upregulated in the serum of human stroke patients compared to controls [65].

Most of the downregulated genes in EHM horses were known equine miRNAs, though not much is currently understood about their biological function in horses. In addition, while miRNAs may play a role in many biological processes, current knowledge of the functions for specific miRNAs is limited to those that have been elucidated from experimental investigation for particular diseases of interest. Thus, there is much unknown regarding the function of many miRNAs, particularly in horses. The human or murine orthologs of most of our downregulated miRNAs (*hsa-miR-199a-3p* [66,67], *hsa-miR-34a-5p* [67,68], *hsa-miR-542-5p* [69,70], *hsa-miR-10a-5p* [71], *hsa-miR-328-3p* [72,73], and hsa-miR-138-5p [74]) have been shown to be involved in cell cycle control, apoptosis, or cancer, with the majority having a tumor suppressor function. In horses, many of these (*eca-miR-138*, *eca-miR-328*, *eca-miR-10b*, *eca-miR-34*, and *eca-miR-199*) have also been identified in the male reproductive tract and are presumed to be involved in cell motility and viability of equine spermatozoa [75]. Additionally, *eca-miR-34c* has been shown to also be downregulated in equine ocular squamous cell carcinoma tissue, and it is presumed that the downregulation promotes tumorigenesis by providing a metabolic advantage through fatty acid synthesis [76]. We also found *eca-miR-146a/eca-miR-146a* downregulated in EHM horses compared to non-EHM horses. In humans, *hsa-miR-146a-5p* targets factors to reduce p38/JNK mediated inflammation in adipocytes [77], and in horses *eca-miR-146a* has been identified in the male reproductive tract [75]. Finally, *eca-miR-328* and *eca-miR-483* have been previously shown to be differentially expressed in the serum of certain breeds, which is interesting considering pony breeds are less likely to acquire EHM compared to other breeds [1,78]. In the serum of ponies, *eca-miR-328* was shown to be downregulated and *eca-miR-483* upregulated compared to the warmblood breed horses [78]. In our study, both of these miRNAs were downregulated in EHM horses compared to non-EHM horses.

Target gene prediction of the differentially expressed miRNAs identified a number of genes that were also identified as differentially expressed genes from mRNA sequencing in the contrast (between group) comparison and/or the within group comparison unique to EHM horses in response to infection (Figure 13, Table 3).

For the predicted target genes of upregulated miRNAs we identified 20 genes common with differentially expressed genes in the contrast (between group) comparison and 51 genes that were common with the differentially expressed genes unique to EHM horses (within group comparison). For the predicted genes of upregulated miRNA, we identified 39 genes common with differentially expressed genes in the contrast comparison and 87 genes that were common with the differentially expressed genes unique to EHM horses. In addition, while there was no miRNA differentially expressed in non-EHM horses, a few of the miRNAs differentially expressed in EHM horses targeted genes that were differentially regulated in non-EHM horses (*SCD*, *CILP*, *LZTS1*, *SHISA5*, *TCF7L1*).

Common predicted target genes of upregulated miRNAs included genes known to be involved in the regulation of the mitogen-activated protein kinase cascades (such as the extracellular signal-related kinase cascade) (*MAPK13*, *FOSB*, *JUND*, *CREB5*, *ATF3*), which are important in T-cell activation and proliferation [59] as well as genes important in the interferon response (*ADAR*, *OAS21*). In addition, there were a number of common predicted target genes with a role in cell migration and cell-to-cell junction assembly (*NTNG2*, *NEO1*, *JAM2*, *F11R*, *PARDG6*, *PTPRF*, *NFASC*, *SDC3*, *PLEKHG5*, *HIC1*), as well as genes with semaphoring and ephrin receptor activity (*PLXNP1*, *SE-MA6C*, *SEMA4C*, *SRGAP1*), which are known to play a role in immune pathologies and neurodegenerative diseases [79].

Common predicted target genes of downregulated miRNAs also included JUN and FOS genes known to be involved in the regulation of the mitogen-activated protein kinase cascades similar to what was observed for the differentially upregulated miRNAs (*DUSP1*, *FOSB*, *JUND*, *CREB5*, *FOS*). In addition, common predicted targets included genes related to the fibronectin III superfamily (*EPHA4*, *FN1*, *PTPRF*, *INSR*, *INSRR*, *NFASC*). These genes are thought to be important for cell adhesion, cytoskeletal reorganization, cell migration and hemostasis. Finally, common predicted targets included kruppel-like-transcription factors (*KLF6*, *KLF4* and *ZBTB16*), which have recently been shown to be involved in DMSO-induced reactivation from latency for bovine herpesvirus-1 [80].

## 4. Discussion

In this experiment we utilized the “old mare model” (female horses > 18 years old) to reliably induce EHM and to compare infection with EHV-1 between EHM and non-EHM “protected” horses. This approach was based on the fact that EHM typically only occurs sporadically in EHV-1 infected horses and is challenging to induce experimentally [19]. However, EHM has been shown to occur in >70% of experimentally infected aged horses and is more likely to occur in mares [13,31]. The increased propensity of older horses to develop EHM following EHV-1 infection is presumed to be a result of differing host immunity. Our goal was to take advantage of this phenomenon and use RNA sequencing to analyze the transcriptomic profile of PBMCs in horses that did and did not develop EHM to elucidate potential host mechanisms involved in the pathogenesis of EHM.

Evaluating this comparison, we found an upregulation of the gene encoding the interleukin-6 (IL-6) protein in PBMCs of EHM horses when compared to non-EHM horses. Excessive IL-6 production is a key feature of the condition known as “cytokine storm” and is considered to be a major contributor to vascular damage during disorders such as sepsis or systemic inflammatory response syndrome [57]. In PBMCs, monocytes and T-cells are the likely sources of IL-6 during an infection [81]. IL-6 is also widely known for its role in immunopathology of several diseases including stroke [56]. Ischemic stroke in humans shares many characteristics with the pathogenesis of EHM, namely the damage to the blood−brain barrier (BBB) resulting in leukocytic inflammation and damage to the CNS. Serum IL-6 routinely predicts the severity of the CNS lesions as well as the clinical outcome in ischemic stroke patients [56,82,83,84,85]. IL-6 is thought to contribute to BBB damage in multiple ways. More specifically, IL-6 can act on endothelial cells to increase their expression of adhesion factors, which contribute to leukocyte adhesion, infiltration, and damage to the BBB [86]. Furthermore, IL-6 also plays a role in promoting thrombosis, another critical feature of BBB damage and neuropathology [87]. In horses, IL-6 has been implicated in equine diseases including equine metabolic syndrome and osteoarthritis, both of which tend to occur in older animals [88,89]. Due to its relevance in vascular damage and immune mediated disease, in addition to the upregulation we observed in the present study, IL-6 may be a candidate for use as a biomarker to predict EHM severity in EHV-1 infected horses.

Consistent with the fact that IL-6 is known to play a role in thrombosis and neu-ropathology [87], we also identified a number of genes related to fibrin clot formation and changes in vascular permeability to be significantly downregulated in EHM horses. Physiologically, hemostasis is tightly regulated by the body, as dysfunction in either timely coagulation or fibrinolysis can result in pathogenic hemorrhage or clotting disorders [90]. In horses, the coagulation cascade is known to be induced during EHV-1 viremia, and it is thought that this is a major contributor to endothelial damage and the immunopathology that leads to EHM [25,26,27,28]. In this study, we found two genes related to fibrin clot formation (*THBS1* and *FN1*) as well as *MMP9*, which is associated with increasing permeability of blood vessels, to be downregulated in PBMCs of EHM horses compared to non-EHM horses. Previously, we also found upregulation of genes encoding proteins involved in coagulation during EHV-1 viremia in PBMCs [91]. In humans, decreases of *THBS1* expression by endothelial cells infected with hantavirus is linked to hemorrhagic disease, and elevated serum levels have been seen in stroke patients [92,93]. Elevated levels of *FN1* and *MMP9* predicts endothelial damage and hemorrhage after stroke in response to thrombolytic therapy [94]. While it is unclear what precise role the downregulation of these genes in PBMCs have in the pathogenesis of EHM, our results further support a role of hemostasis (dys)regulation in EHM.

We also observed an upregulation of several interferon stimulated genes (ISGs) in EHM horses. These genes tend typically to be related to the innate immune response to viruses. Many of these ISGs were upregulated in both EHM and non-EHM horses, which is in agreement with our previous observations [91]. However, EHM horses appeared to have more significantly upregulated genes related to the antiviral defense and type 1 interferon response than the non-EHM group, such as the pattern recognition receptor, *TLR3*, the interferon regulatory factors *IRF7* and *IRF9*, and interferon stimulated genes, such as *RSAD2*. An early and rapid induction of interferons has been shown to be critical for protection from viral diseases and we have previously demonstrated significantly lower amounts of IFNα protein in the nasal secretions of horses that went on to develop EHM when compared to horses that did not develop EHM at 24 h after EHV-1 CH infection [14]. Confirming this observation, the EHM horses in this study also exhibited significantly lower IFNα secretion in the nasal secretion at 24 and 48 h post infection when compared to the non-EHM horses [54]. In viral infection, the timing of type 1 interferon pathway induction in relation to peak viral replication can dictate the severity of downstream immunopathology and disease, because delayed interferon responses to increased viral titers contribute to immunopathology, presumably through an exaggerated proinflammatory cytokine response and leukocytic activation [95,96,97]. The present study does not provide information on the timing of the interferon response or whether there is a delay in induction of interferon response in PBMCs resulting from the lower interferon levels in the nasal secretions and the reduced nasal shedding early on in EHM horses. There also is no information on how this contributes to the propensity for the development of high-er viremia and EHM. However, the higher viremia levels in EHM horses likely explain the increased defense response genes observed in this group and could contribute to the downstream immunopathology of the CNS vasculature. Future work should focus on the expression patterns of the interferon associated genes over time.

While IL-6 and associated inflammatory factors likely play a major role in the immunopathology of the CNS, our results suggest that the dysregulation of the development of an appropriate T-cell response also appears to be involved in the likelihood of developing EHM. In the current study, *FOS*, the gene encoding the c-fos protein and the direct product of the ERK cascade, was downregulated as well as the gene encoding the other AP-1 subunit, *JUN*. Many genes associated with or influenced by the MAPK/ERK cascades also appeared to be differentially regulated in EHM horses and in addition many of the differentially regulated miRNAs identified in EHM horses have predicted target genes associated with or influenced by the MAPK/ERK cascades. Previously, in vitro experiments have shown that EHV-1 infection of PBMCs stimulates the MAPK pathway in infected cells, and thus enhances cell adhesion and viral trans-fer with the vascular endothelial cells [98,99]. In our study, we focused on the entire PBMC population, not only infected cells. The differential expression of MAPK pathways in the PBMCs of EHM horses points to the importance of this pathway in the pathogenesis of EHM from an immunological perspective. Taken together with the literature on immunosenescence, our gene expression data support the idea that impairment of T-cell activation and cellular immunity plays an important role in EHM pathogenesis. Increased age has been identified as a risk factor for the development of EHM [13,29]. This was confirmed in our study where EHM developed in 7/7 aged horses and only 1/7 young horses. Immunosenescence is a well-known phenomenon affecting aged individuals, with T-cells being considered the most affected immune cell population [100,101]. One of the key features of immunosenescence is the resistance of T-cells to activate and proliferate upon antigenic stimulation [32]. Ligand/receptor binding (such as T-cell receptor stimulation by antigen) triggers a variety of signal transduction pathways, which ultimately result in transcription factor production. Interleukin-2 (IL-2) expression is the hallmark feature of T-cell activation and proliferation. The binding of several transcription factors to the IL-2 promotor, including the transcription factor AP-1 are required for full activation of IL-2 production [59]. The signal transduction pathway responsible for AP-1 production is the mitogen activated protein kinase (MAPK) cascade. This cascade involves the production of extracellular signal-related kinase (ERK), which then enters the nucleus to facilitate transcription of the FOS gene and phosphorylation of the c-fos protein. The c-fos and c-jun proteins form the subunits of AP-1, the transcription factor involved in IL-2 production [59]. Activity of AP-1 has been shown to be impaired in aged mice, and this is related to a decrease in FOS expression [102]. In humans, age-related impairments in AP-1 activity have also been shown to be associated with decreased IL-2 production in T-cells [103]. Aging is known to reduce MAPK activation and the ERK1/ERK2 cascade [103,104,105]. Further, impairment of the activation of the ERK pathway is known to be associated with decreased IL-2 production by T-cells in response to stimulation in aged humans [106]. Finally, age-related immunosenescence is also known to be associated with a decreased lymphoproliferative response in humans [107]. In horses, it has been shown that PBMCs from aged horses exhibit a reduced lymphoproliferative response in vitro when exposed to mitogens compared to younger horses [108,109,110]. In our study, there was a significant upregulation of CD4+ memory cell activation in non-EHM horses, which is a response we did not observe in EHM horses. Furthermore, we have previously shown that lymphopenia, as well as decreased T-cell responses are part of the immune response to EHV-1 infection, even in horses that do not develop EHM [34,111,112]. Consistent with this, our data indicate there is a downregulation of AP-1 production (through *FOS/JUN* downregulation) in horses that develop EHM when compared to non-EHM horses, most likely through dysregulation of MAPK/ERK cascades.

In persistent viral infections (as occur with many herpesviruses), reactivation is suppressed by constant immune surveillance. It has been suggested for human herpesviruses that this chronic antigenic stimulation contributes to immunosenescence, specifically in CD4+ and CD8+ memory T-cells [113]. This is thought to be because after a lifetime of replication, the memory cell population enters a late stage of effector differentiation and senescence [32,114]. It appears different herpesviruses may affect this in different ways. For example, varicella-zoster virus specific CD4+ T-cells are shown to decline with age, while human cytomegalovirus specific T-cell clones can grow to become a prominent proportion of the memory T-cells [32,115,116]. Nevertheless, frequent antigenic stimulation throughout the life of the horse likely influences the EHV-1-specific T-cell population in aged horses, though more work is needed to verify this.

In addition to the dysregulation of MAPK cascades, *JUN/FOS* expression, and the presumed decrease in T-cell activation, we also observed an apparent skew towards a T-helper type 2 (Th2) immune response in the EHM horses when compared to the non-EHM group. The T-helper type 1 (Th1) cytokine genes *IFNG* and *TBX21* were downregulated in EHM horses. Though many phenotypes of CD4+ T-cells have been identified, the classic dichotomy dictates that CD4+ T-cells express a repertoire of cytokines that fit either a Th1 (proinflammatory) or Th2 (anti-inflammatory) upon activation. Th1 immunity contributes to the effective clearance of viral infections and encourages CTL responses, which are the only known correlate of protection from EHV-1 [13,117,118]. In further support of a transition to a Th2 immune response and its contribution to EHM, the horses that developed EHM in our study showed higher levels of IL-10 in nasal secretions and CSF and lower IFNα and IL-17 in nasal secretions when compared to non-EHM horses [54]. Furthermore, EHM horses induced a greater EHV-1 IgG(T) antibody subisotype response following EHV-1 CH compared to non-EHM horses [54]. This subisotype has been associated with a Th2 immune response in horses [119]. The observed skew from Th1 to Th2 immunity in horses that develop EHM indicates that proper Th1 mediated immune responses are important protective features against EHV-1 infection and EHM and a shift to Th2 responses may predispose horses to clinical EHM. Further supporting the skew toward a Th2 response in horses that are effected by EHM is previous data from our laboratory showing that young horses with clinical EHM also showed decreased IFNα but elevated IL-10 in nasal secretions [14].

We also saw some differential expression of chemokines in EHM horses compared to non-EHM horses. *CXCL10* is best known for its T-cell chemotactic activity and its receptor can be found on Th1 cells, CD8+ cytotoxic lymphocytes, and NK cells [120]. We observed an upregulation of *CXCL10* in EHM horses, but not in non-EHM horses. This is in contrast to our previous study where we found a statistically significant in-crease in *CXCL10* expression in PBMCs following EHV-1 CH [91]. Additionally, infection of PBMCs in vitro has been shown to increase *CXCL10* expression [121]. Thus, it was surprising to see that *CXCL10* was only significantly upregulated in the EHM horses in this study. However, upon further examination, it was apparent that *CXCL10* was upregulated in the non-EHM horses in this study as well, but this difference was not statistically significant (FDR = 0.16, log fold change = 1.5). Additionally, we saw that *CCL5* was downregulated in EHM horses. Known as “regulated upon activation normal T-cell expressed and secreted” or “RANTES”, *CCL5* acts as a chemotactic agent for monocytes, T-cells, NK cells, dendritic cells, eosinophils, and basophils [122]. Aging may have an impact on the sensitivity of PBMCs to secrete *CCL5* following stimulation. In human PBMCs, T-cells from the elderly produced more *CCL5*, while NK cells produced less when compared to PBMCs from younger people [123]. While the role of the downregulation of *CCL5* from PBMCs is unknown in EHM pathogenesis, it likely reflects differences in cell populations, activation status, and responsiveness between these two groups.

In addition to host transcripts, we were able to identify viral transcription in the PBMCs using RNA sequencing analysis. EHV-1 transcription was detected in all samples post EHV-1 CH, but no noticeable differences in gene expression between EHM and non-EHM horses were observed. We found very little EHV-1 transcription in PBMCs prior to EHV-1 CH, and only two pre-CH samples (0/6 non-EHM horses and 2/8 EHM horses) were positive for EHV-1 transcripts. This is in contrast to our previous study, where we found low-level transcription in five out of seven horses prior to EHV-1 CH, which likely indicated latent infection [91]. It is possible that the previously described horses were latently infected with EHV-1, while the majority of the horses in the present study were not. However, this is unlikely as most horses are exposed to EHV-1 at a young age, often before 1 year of age [124,125,126,127]. Another explanation could be differences in the sensitivities of the two analyses. However, while the RNA quality was slightly better in our previous study, the depth of sequencing and mapping quality was comparable between the two studies. Notably, the EHV-1 transcription profile analysis can only reflect the circulating latently infected PBMC pool at the moment of sampling. Latently infected PBMCs may recirculate between lymphatic tissue and the blood circulation, which could therefore explain the different findings prior to CH. Another explanation is that there is a difference in the “depth” of latency in PBMCs between the different herds in these experiments. Alpha herpesviruses may exhibit several gene expression profiles during latency and “deeper” latency is characterized by lower transcription and decreased likelihood of reactivation [128,129,130,131]. It is possible different factors between the herds contributed to modulation of EHV-1 depth of latency in the PBMCs. We additionally identified transcripts from the gamma herpesviruses EHV-2 and EHV-5 in several samples both pre and post EHV-1 CH with no apparent effect of EHV-1 CH, which is what we observed previously [91]. We did observe that EHV-2 transcripts were more prevalent in the non-EHM group—which may indicate that younger horses are more likely to have active transcription of EHV-2.

Small RNA sequencing and analysis identified several known and novel host and viral miRNAs expressed in the PBMC samples of these horses. Interestingly, and in agreement with our previous findings [91], no miRNAs mapped to the EHV-1 or EHV-4 genomes, even during peak EHV-1 viremia. In agreement with our previous study, several miRNAs were also identified for the equine gamma herpesviruses, EHV-2 and EHV-5. The gamma herpesviruses are known for their persistence in lymphocytes [132,133]. Given our findings, it is presumed that miRNA expression plays a role in the maintenance of latent lymphocytic infection of gamma herpesviruses. The lack of EHV-1 miRNAs indicate that EHV-1 likely does not make use of miRNA expression during lymphocytic infection during peak viremia. We also identified numerous equine (host) miRNAs expressed by the PBMCs; however, very few of these were differentially expressed as a result of EHV-1 CH. Of the few miRNAs altered during EHV-1 infection, the majority were downregulated in EHM horses during infection. The human and murine orthologs of these miRNAs have routinely been associated with cancer, presumably due to their role in regulation of the cell cycle [66,67,68,69,70,71,72,73,74]. Most notably, predicted target genes of the identified differentially regulated miRNAs in our study directly targeted a large number of genes that were also shown to be differentially regulated in EHM horses in response to infection. Not surprisingly many of the targeted genes were involved with the regulation of MAPK/ERK cascades, stimulation of interferons, hemostasis, cytoskeletal reorganization and cell migration and thus further support our RNA expression data.

For this study, we used the “old mare model” in order to reliably induce EHM in horses using EHV-1 CH infection. It has been well established that aged female horses are significantly more likely to develop EHM during acute EHV-1 infection, and we also observed this in our study where all aged (but only one young) horses developed EHM, all of which were female [13,19,29]. It is known that factors associated with age and sex play a role in the development of EHM [1,13]. It is also known that EHV-1 associated abortions are rare when infection occurs in early pregnancy, but more frequent when it occurs in late pregnancy. It is presumed that changes to the hormonal microenvironment of the pregnant uterus predisposes this site to vasculitis and thrombosis at different stages of pregnancy [134]. In our study, we observed genes associated with the response to progesterone upregulated in EHM horses following EHV-1 CH. It was not within the scope of this study to tease out specific differences between ages and sexes of horses, but rather to reliably induce EHM. More work is needed to study factors in young and male horses that develop EHM. Additionally, focusing on more timepoints looking at early immune events in the PBMCs and the respiratory tract on day 1 or 2 following infection could help to further elucidate host factors important for protection from EHM.

In summary, the features we found to be associated with EHM such as elevated IL-6 expression and decreased T-cell activation through reduced AP-1 production can be explained with the phenomena associated with the aging immune system. Regardless, these mechanisms are also likely involved in EHM development in younger horses and were observed in the young horse exhibiting EHM in our study as well. Further host mechanisms identified to be associated with EHM included factors involved in coagulation, cell adhesion, and the response to progesterone. Future studies should follow up on these factors identified as risk factors (or protective factors) for EHM in cohorts of horses of all age groups and of both sexes as well as in different breeds. In conclusion, this study provides an unbiased insight into differences in systemic responses during peak viremia of horses with and without EHM during EHV-1 infection and identifies inhibition of IL-6, regulation of hemostasis, mechanisms to activate T-cells, and shifting immune responses toward Th1 cell-mediated immunity as interesting targets for protection from EHM.

## Figures and Tables

**Figure 1 viruses-13-00356-f001:**
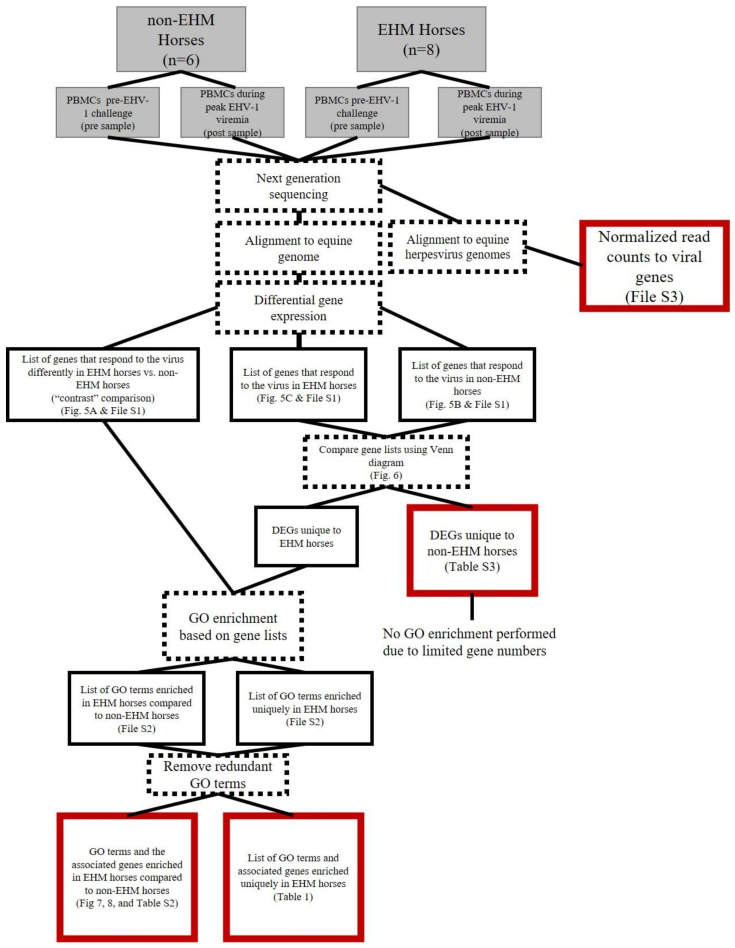
Summary of mRNA sequencing data analysis. RNA was extracted from PBMCs from horses prior to and during EHV-1 CH. The gray boxes represent experimental design. The dashed outline boxes are data analysis processes (actions). Black and red boxes represent output. The red outline indicates final output used for interpretation.

**Figure 2 viruses-13-00356-f002:**
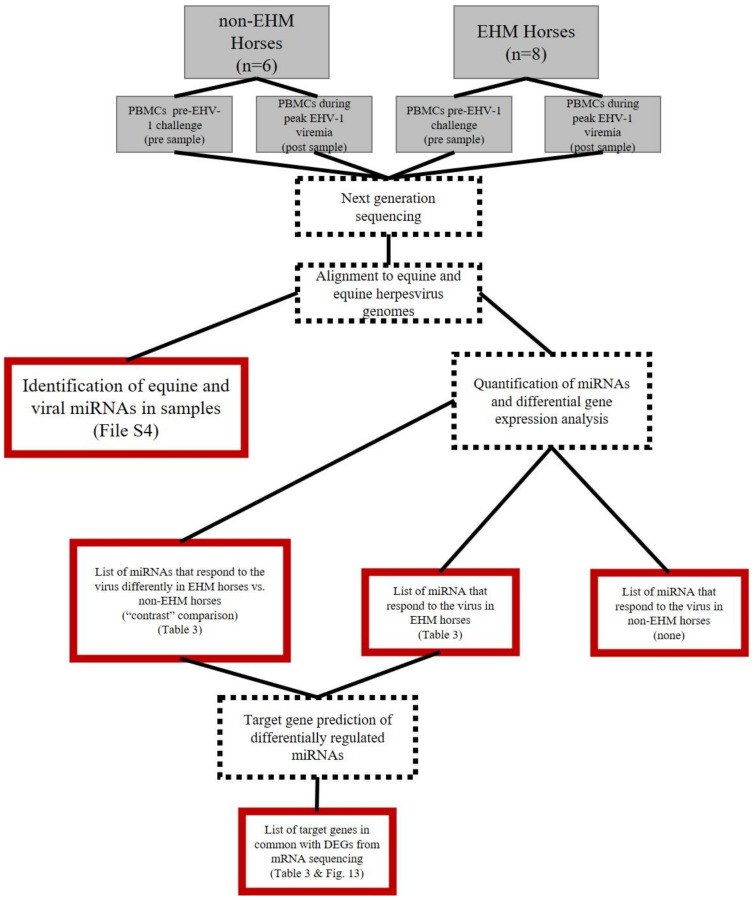
Summary of miRNA sequencing data analysis. RNA was extracted from PBMCs from horses prior to and during EHV-1 CH. The gray boxes represent experimental design. The dashed outline boxes are data analysis processes (actions). Black and red boxes represent output. The red outline indicates final output used for interpretation.

**Figure 3 viruses-13-00356-f003:**
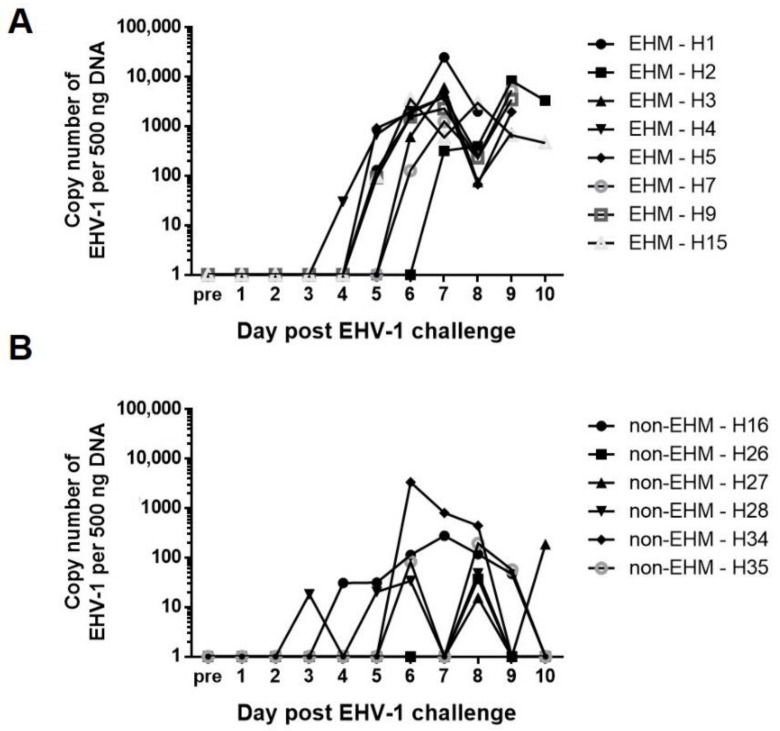
Viremia in (**A**) EHM and (**B**) non-EHM horses. Data is expressed as EHV-1 copy number per 500 ng template DNA as determined by qPCR. This data was obtained in conjunction with another study [54]. Day of peak viremia was determined for each horse and PBMCs from that day were used for each horse’s post-EHV-1 CH sample.

**Figure 4 viruses-13-00356-f004:**
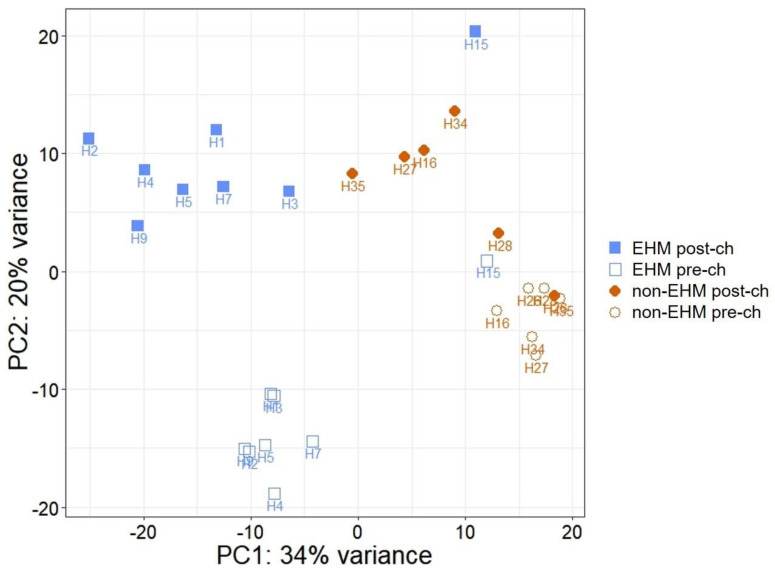
Principle component analysis (PCA) plot. Principle component analysis was performed on the read count data obtained after mRNA sequencing. Brown closed diamonds indicate samples from non-EHM horses post EHV-1 CH, brown open circles indicate samples from non-EHM horses prior to EHV-1 CH, blue closed squares indicate samples from EHM horses post EHV-1 CH, and blue open squares indicate samples from EHM horses prior to EHV-1 CH.

**Figure 5 viruses-13-00356-f005:**
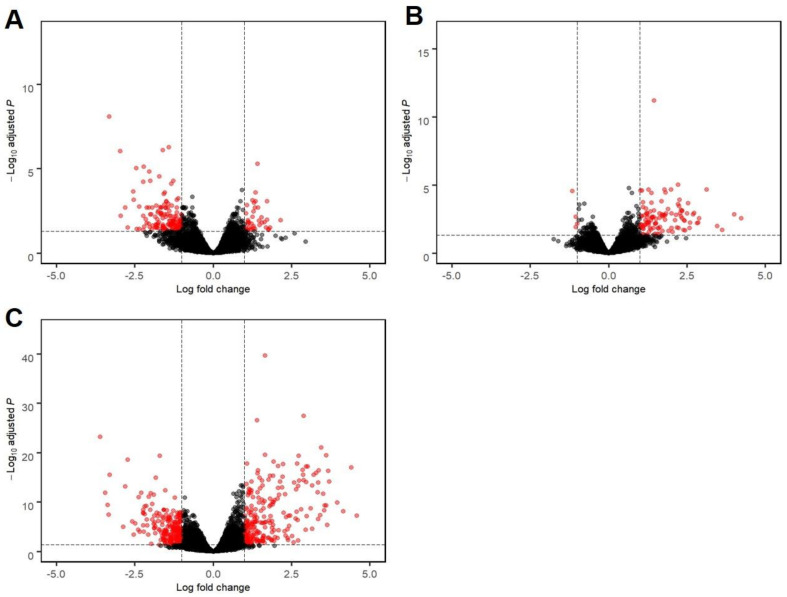
Volcano plots of differentially expressed genes. (**A**) Genes differentially expressed between EHM and non-EHM horses. Genes with a positive log fold change represent genes upregulated in EHM horses compared to non-EHM horses, while genes with a negative log fold change represent genes downregulated in EHM horses compared to non-EHM horses. (**B**) Genes differentially expressed within non-EHM horses. Genes with a positive log fold change represent genes upregulated in non-EHM horses post CH compared to pre CH, while genes with a negative log fold change represent genes downregulated. (**C**) Genes differentially expressed within EHM horses. Genes with a positive log fold change represent genes upregulated in EHM horses post CH compared to pre CH, while genes with a negative log fold change represent downregulated genes. *p* values are expressed on the *y*-axis, with more significantly differentially expressed genes towards the top of the plot. Genes highlighted in red passed the threshold of significance set at adjusted *p*-value < 0.05 and log fold change >|1|.

**Figure 6 viruses-13-00356-f006:**
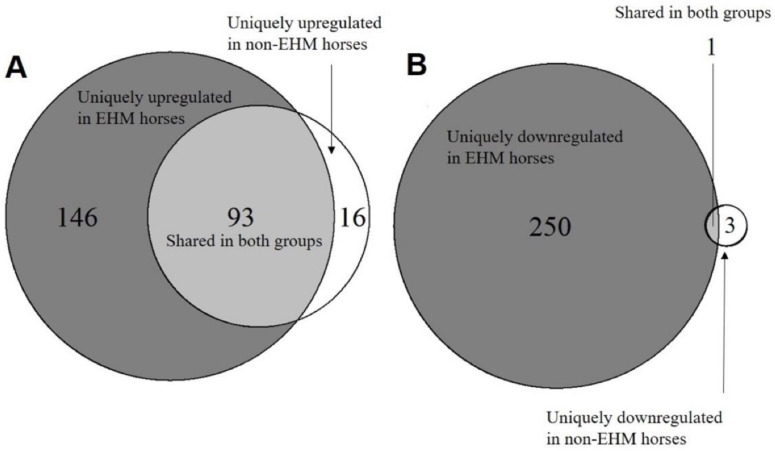
Venn diagrams of up- and downregulated genes in response to EHV-1 challenge. The within group comparisons for EHM and non-EHM horses were compared using a Venn diagram to determine genes uniquely (**A**) upregulated and (**B**) downregulated for each group.

**Figure 7 viruses-13-00356-f007:**
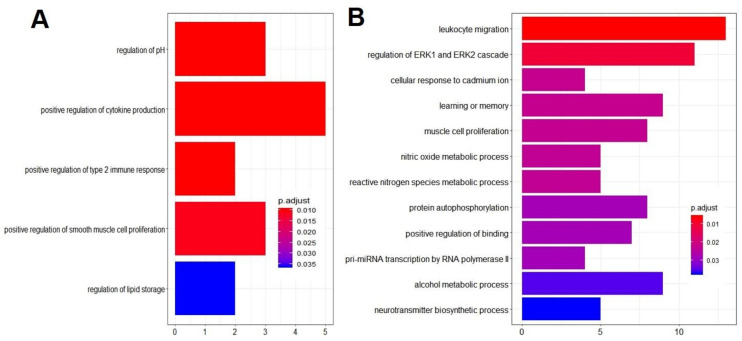
GO terms for biological processes overrepresented in EHM horses compared to non-EHM horses. (**A**) Upregulated biological processes and (**B**) downregulated biological processes. GO term enrichment analysis was performed using the enrichgo function of the clusterprofiler package in R. The resulting terms were filtered for redundancy using REVIGO. The nonredundant enriched GO terms are visualized here. The most significantly enriched terms are at the top and listed in decreasing significance (increasing p.adjust). The number of genes from our gene list are indicated on the *x*-axis.

**Figure 8 viruses-13-00356-f008:**
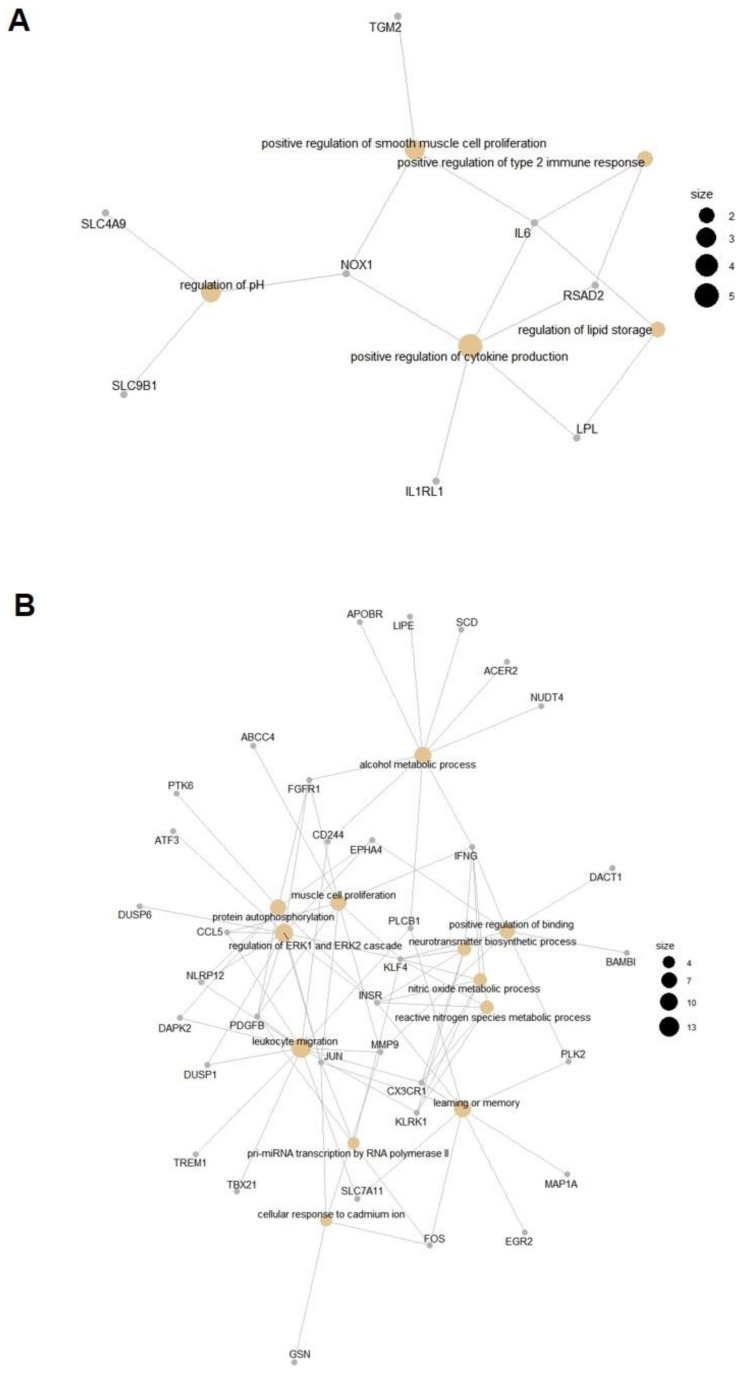
Net plot of the most significantly enriched GO terms and associated genes. (**A**) Upregulated biological process and (**B**) downregulated biological processes. The nonredundant GO terms are listed here with the associated genes from our gene list. Tan nodes represent the GO term and gray nodes represent genes. The size of the GO term nodes indicates the number of genes from our list associated with that term. The biological processes and the associated genes cluster based on similarity.

**Figure 9 viruses-13-00356-f009:**
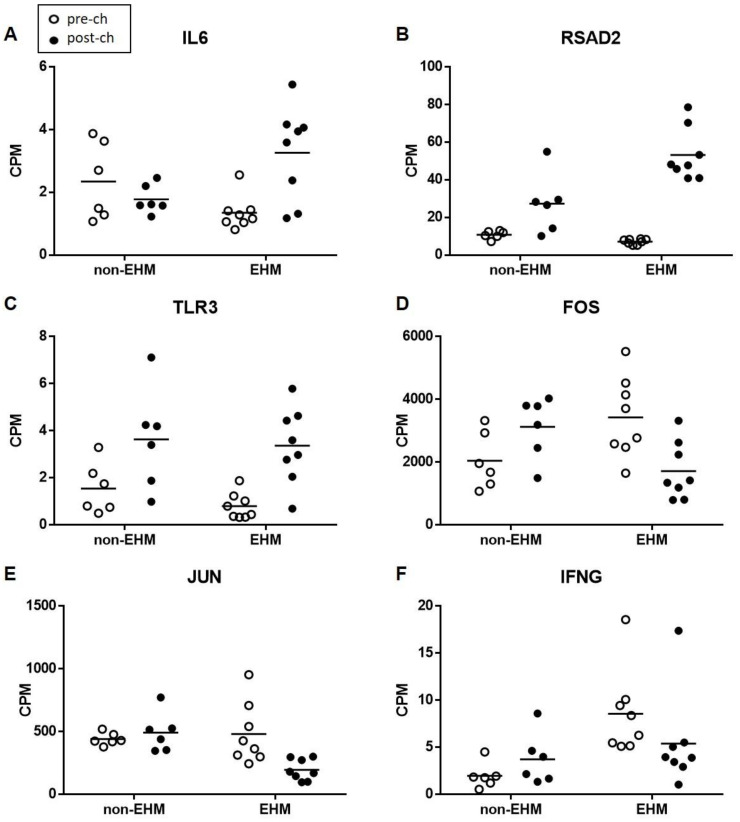
Normalized read counts (cpm) of selected genes in PBMCs of horses. (**A**) IL6 (**B**) RSAD2 (**C**) TLR3 (**D**) FOS (**E**) JUN (**F**) IFNG. All differences in responses are statistically significant at *p* < 0.05.

**Figure 10 viruses-13-00356-f010:**
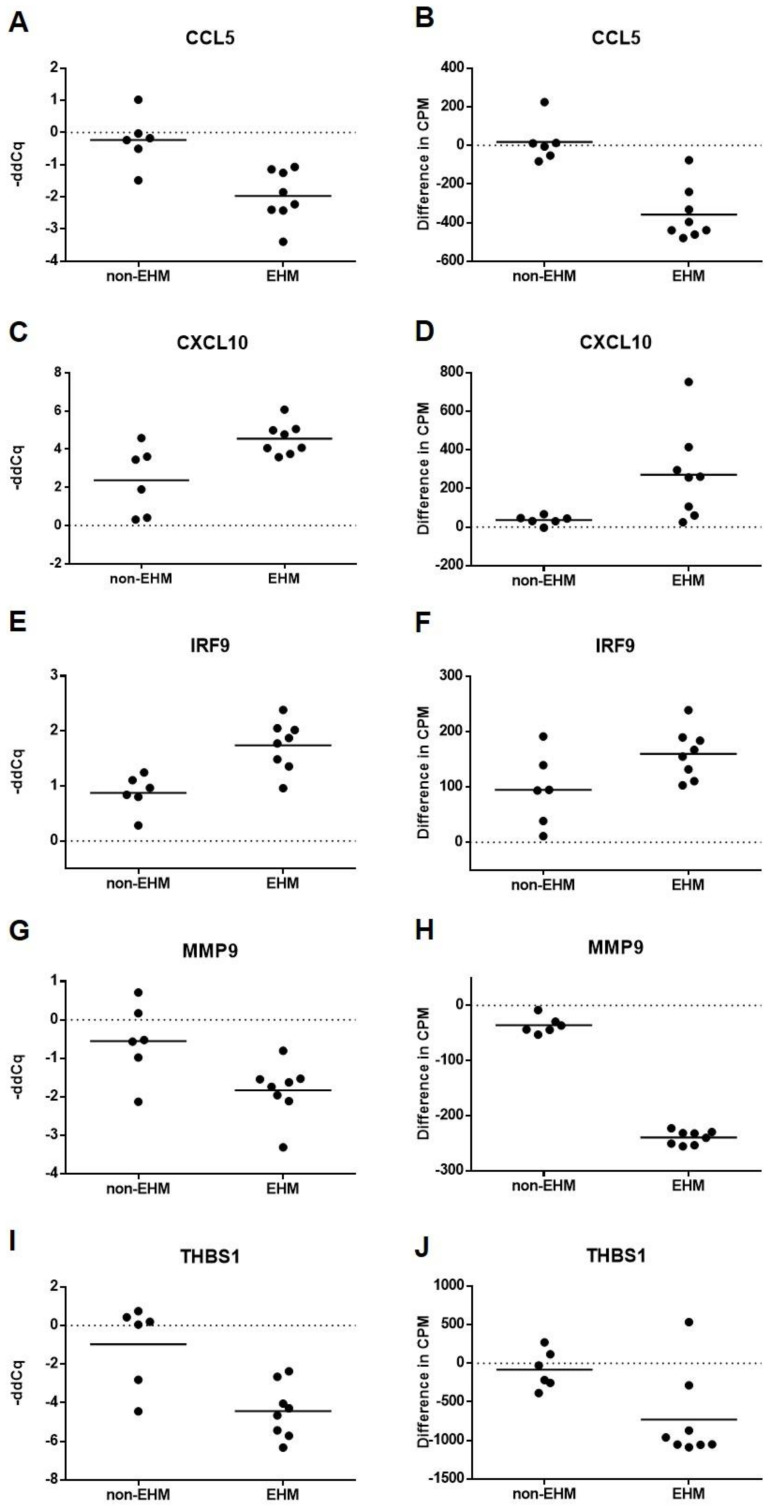
Gene expression as determined by RT-qPCR and RNA sequencing. RT-qPCR data from whole blood is expressed as the negative delta-delta-Cq value for (**A**) CCL5, (**C**) CXCL10, (**E**) IRF9, (**G**) MMP9, and (**I**) THBS1, where the average pre CH data for each group used as a calibrator and values above zero represent an upregulation and below zero represent a downregulation. RNA sequencing data from PBMCs are shown for comparison and expressed as delta-CPM (post CH CPM value—group average of pre CH CPM) for (**B**) CCL5, (**D**) CXCL10, (**F**) IRF9, (**H**) MMP9, and (**J**) THBS1. Differences between groups were statistically significant as described in the methods for all genes visualized here.

**Figure 11 viruses-13-00356-f011:**
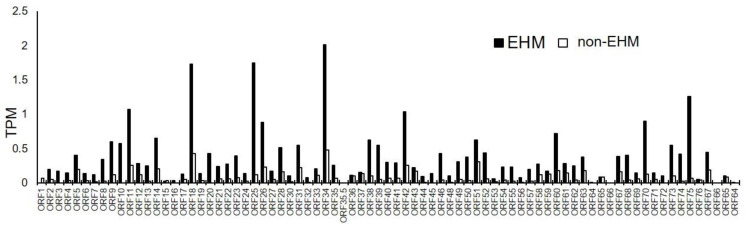
Normalized counts of viral genes post EHV-1 challenge. Data represents the average transcripts per million (TPM). The gray line is non-EHM horses and the black line is EHM horses.

**Figure 12 viruses-13-00356-f012:**
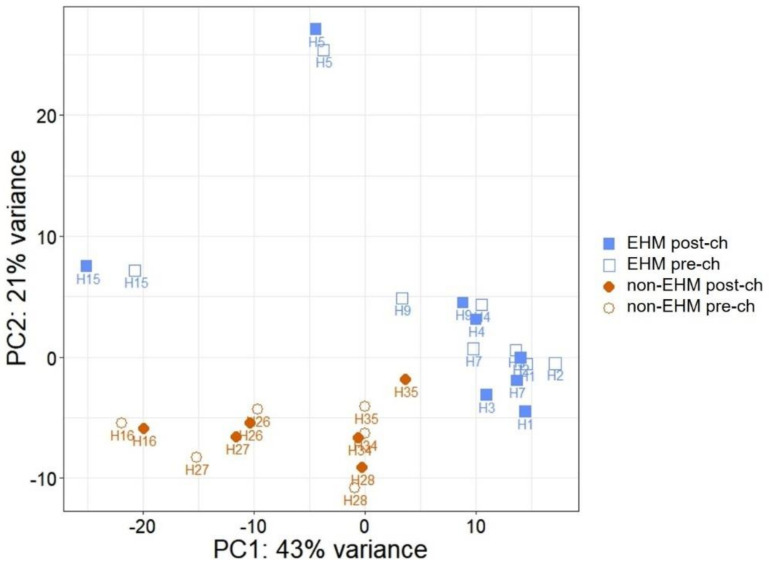
Micro RNA expression analyses. Principal component analysis (PCA) plot for miRNA read counts. Principal component analysis was performed on the read count data obtained after miRNA sequencing. Brown closed diamonds indicate samples from non-EHM horses post EHV-1 CH, brown open circles indicate samples from non-EHM horses prior to EHV-1 CH, blue closed squares indicate samples from EHM horses post EHV-1 CH, and blue open squares indicate sam-ples from EHM horses prior to EHV-1 CH.

**Figure 13 viruses-13-00356-f013:**
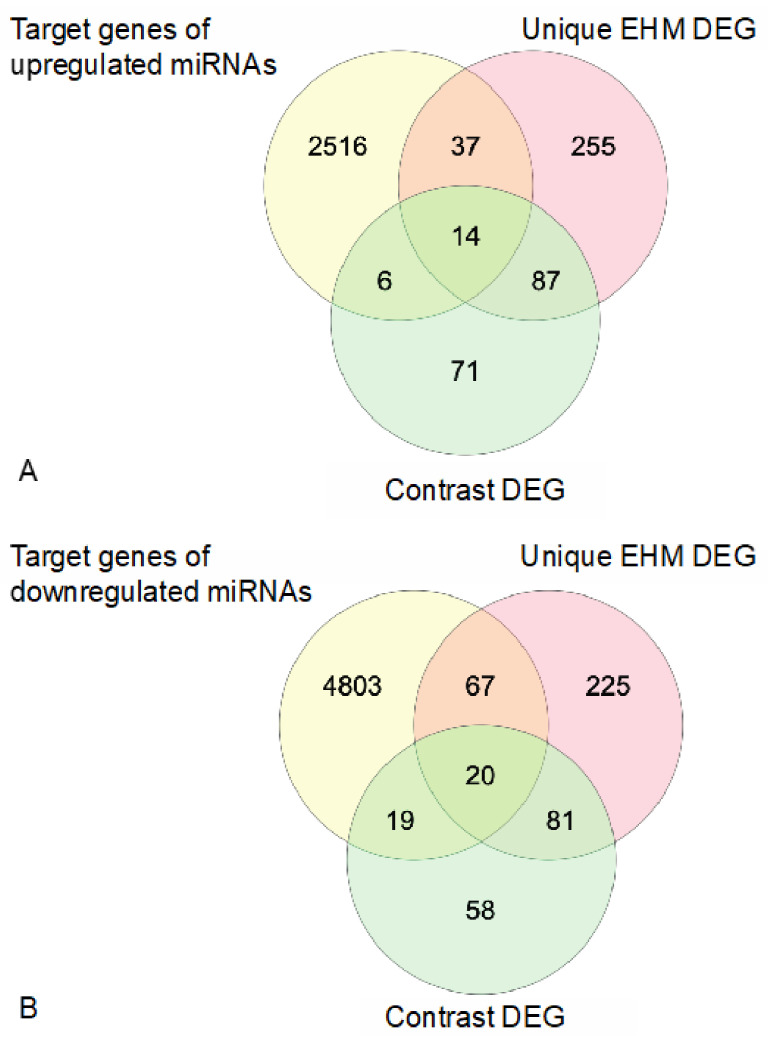
Venn diagrams highlighting genes identified as miRNA targets that were common to those differentially regulated in response to EHV-1 infection. (**A**) Predicted target genes of differ-entially upregulated miRNA that are common with differentially expressed genes unique to EHM horses (within group comparison) and/or differentially expressed genes in the contrast (between group comparison). (**B**) Predicted target genes of differentially downregulated miRNA that are common with differentially expressed genes unique to EHM horses (within group comparison) and/or differentially expressed genes in the contrast (between group) comparison.

**Table 1 viruses-13-00356-t001:** Enriched GO terms based on differentially expressed genes in EHM horses. GO term enrichment was performed on the gene list of within group up- and down-regulated genes unique to EHM horses.

ID	Description	Gene Ratio	Bg Ratio	*p* Value	p.adjust	q Value	geneID	Count
**Upregulated**								
GO:0051607	defense response to virus	13/92	175/13,991	1.1 × 10^−10^	2.1 × 10^−07^	1.8 × 10^−07^	EIF2AK2/TRIM56/ADAR/CXCL10/IRF9/RTP4/ISG15/AIM2/IRF7/TLR3/IL6/ISG20/IL15	13
GO:0032479	regulation of type I interferon production	8/92	104/13,991	3.9 × 10^−07^	1.3 × 10^−04^	1.1 × 10^−04^	TRIM56/UBA7/NMI/ISG15/IRF7/ACOD1/ TLR3/ZBP1	8
GO:0042107	cytokine metabolic process	7/92	101/13,991	4.4 × 10^−06^	6.8 × 10^−04^	5.9 × 10^−04^	LAG3/NMI/IGF2BP3/IRF7/TLR3/IL6/CYBB	7
GO:0043900	regulation of multi-organism process	11/92	313/13,991	6.3 × 10^−06^	9.0 × 10^−04^	7.9 × 10^−04^	EIF2AK2/ADAR/PPID/ISG15/CD180/AIM2/ ACOD1/ISG20/IL15/SLPI/TIMP1	11
GO:0048771	tissue remodeling	7/92	150/13,991	5.8 × 10^−05^	4.5 × 10^−03^	3.9 × 10^−03^	TGM2/CD38/SNX10/IL6/IL15/VDR/TIMP1	7
GO:0032570	response to progesterone	4/92	38/13,991	1.1 × 10^−04^	6.7 × 10^−03^	5.9 × 10^−03^	CD38/ACOD1/NR1H3/RAMP2	4
GO:0010883	regulation of lipid storage	4/92	41/13,991	1.5 × 10^−04^	7.9 × 10^−03^	6.9 × 10^−03^	IL6/LPL/NR1H3/MSR1	4
GO:0015012	heparan sulfate proteoglycan biosynthetic process	3/92	23/13,991	4.4 × 10^−04^	1.8 × 10^−02^	1.6 × 10^−02^	EXT1/TCF7L2/EXTL2	3
GO:0010743	regulation of macrophage derived foam cell differentiation	3/92	25/13,991	5.7 × 10^−04^	2.1 × 10^−02^	1.9 × 10^−02^	LPL/NR1H3/MSR1	3
GO:0070661	leukocyte proliferation	7/92	231/13,991	8.1 × 10^−04^	2.6 × 10^−02^	2.2 × 10^−02^	TNFSF13B/CD38/CD180/IL6/GAPT/IL15/CCL8	7
GO:0090077	foam cell differentiation	3/92	30/13,991	9.8 × 10^−04^	2.8 × 10^−02^	2.4 × 10^−02^	LPL/NR1H3/MSR1	3
GO:0097050	type B pancreatic cell apoptotic process	2/92	10/13,991	1.9 × 10^−03^	4.7 × 10^−02^	4.1 × 10^−02^	IL6/TCF7L2	2
**Downregulated**								
GO:0030335	positive regulation of cell migration	20/181	441/13,991	1.1 × 10^−06^	8.4 × 10^−04^	7.6 × 10^−04^	DAPK2/MMP9/DOCK5/FGFR1/CASS4/CCL5/FN1/HSPA5/THBS1/JUN/INSR/ADAM8/PDGFB/PDGFD/SEMA6C/SEMA4C/TNFSF14/GATA2/RHOB/GPNMB	20
GO:0031952	regulation of protein autophosphorylation	6/181	43/13,991	1.8 × 10^−05^	8.9 × 10^−03^	8.0 × 10^−03^	NLRP12/JUN/PDGFB/PDGFD/ERRFI1/GPNMB	6
GO:0046777	protein autophosphorylation	12/181	220/13,991	2.9 × 10^−05^	1.3 × 10^−02^	1.1 × 10^−02^	DAPK2/PTK6/FGFR1/NLRP12/JUN/INSR/ PDGFB/PDGFD/ERRFI1/INSRR/BMX/GPNMB	12
GO:0070371	ERK1 and ERK2 cascade	13/181	292/13,991	1.1 × 10^−04^	3.0 × 10^−02^	2.7 × 10^−02^	FGFR1/CCL5/FN1/NLRP12/FBLN1/JUN/INSR/PDGFB/PDGFD/ERRFI1/INSRR/ZFP36L2/ GPNMB	13
GO:0032103	positive regulation of response to external stimulus	12/181	260/13,991	1.5 × 10^−04^	3.6 × 10^−02^	3.3 × 10^−02^	DAPK2/FGFR1/CCL5/FAM19A3/NLRP12/ THBS1/MAPK13/ADAM8/PDGFB/PDGFD/ NPY/TNFSF14	12
GO:0010035	response to inorganic substance	17/181	491/13,991	2.2 × 10^−04^	4.6 × 10^−02^	4.2 × 10^−02^	EEF1A2/MMP9/HSPA5/THBS1/MAPK13/JUN/SELENOP/PDGFD/PTCH1/GSN/FOSB/FOS/ SLC40A1/CHP2/JUND/RHOB/IL1A	17
GO:0071248	cellular response to metal ion	9/181	161/13,991	2.5 × 10^−04^	4.6 × 10^−02^	4.2 × 10^−02^	MMP9/HSPA5/JUN/GSN/FOSB/FOS/SLC40A1/CHP2/JUND	9

**Table 2 viruses-13-00356-t002:** Average fraction of cell populations.

	EHM Pre Challenge (% of Total Cell Population)	EHM Post Challenge (% of Total Cell Population)	Non-EHM Pre Challenge (% of Total Cell Population)	Non-EHM Post Challenge (% of Total Cell Population)
B cells naïve	23.03 ± 1.84	24.75 ± 3.23	37.03 ± 1.53	32.55 ± 1.14 **
B cells memory	0.60 ± 0.60	0.41 ± 0.41	0.03 ± 0.03	0.00 ± 0.00
Plasma cells	0.27 ± 0.12	0.46 ± 0.29	0.28 ± 0.15	0.74 ± 0.06 *
T cells CD8	7.92 ± 1.56	2.36 ± 1.59 *	2.18 ± 0.79	2.21 ± 1.03
T cells CD4 naive	1.64 ± 0.66	4.60 ± 2.05	6.21 ± 2.25	10.26 ± 1.62
T cells CD4 memory resting	0.77 ± 0.55	1.34 ± 0.77	0.02 ± 0.02	0.00 ± 0.00
T cells CD4 memory activated	7.88 ± 0.86	9.55 ± 1.62	0.55 ± 0.26	1.71 ± 0.52 *
T cells follicular helper	16.62 ± 1.58	14.86 ± 1.84	23.27 ± 0.80	19.43 ± 1.44 *
T cells regulatory (Tregs)	1.82 ± 0.62	0.16 ± 0.16 **	1.48 ± 0.49	0.41 ± 0.41
T cells gamma delta	0.88 ± 0.33	2.84 ± 0.66 *	0.79 ± 0.32	2.67 ± 0.82 *
NK cells resting	6.43 ± 1.10	2.75 ± 1.08 **	2.24 ± 0.82	0.86 ± 0.35
NK cells activated	0.13 ± 0.13	0.34 ± 0.22	0.63 ± 0.42	0.53 ± 0.26
Monocytes	12.71 ± 2.75	13.85 ± 3.72	7.27 ± 0.57	11.85 ± 2.22
Macrophages M0	2.99 ± 0.91	0.00 ± 0.00 **	1.77 ± 1.01	0.15 ± 0.15
Macrophages M1	0.00 ± 0.00	0.98 ± 0.30 **	0.00 ± 0.00	0.00 ± 0.00
Macrophages M2	6.34 ± 0.57	8.54 ± 0.80 *	4.22 ± 0.84	5.54 ± 0.93
Dendritic cells resting	0.04 ± 0.04	0.86 ± 0.37 *	0.37 ± 0.29	0.35 ± 0.30
Dendritic cells activated	2.92 ± 0.48	4.88 ± 0.89 *	4.24 ± 0.32	5.99 ± 0.42 *
Mast cells resting	0.33 ± 0.33	0.23 ± 0.22	0.00 ± 0.00	0.00 ± 0.00
Mast cells activated	4.51 ± 0.78	1.80 ± 0.99 *	5.43 ± 2.04	2.32 ± 1.80
Eosinophils	1.63 ± 0.77	3.83 ± 0.58 **	1.74 ± 0.60	1.56 ± 0.50
Neutrophils	0.54 ± 0.23	0.60 ± 0.24	0.25 ± 0.25	0.86 ± 0.56

Table 2. Average fractions of cell populations in PBMCs. Cell population fractions for each sample were estimated using CIBERSORTx [52] and the reference gene signature “LM22” included with the software, which is based on the transcriptome of human PBMC samples with predetermined cell populations. Wilcox signed-rank test was performed on the paired samples for each group * indicates a significant difference at *p* ≤ 0.1, and ** indicates *p* ≤ 0.05 in cell fraction between pre CH and post CH samples for each group (EHM and non-EHM).

**Table 3 viruses-13-00356-t003:** Differentially expressed miRNAs.

mirBase ID	Mouse or Human Orthologue	Mature Sequence	logFC	FDR	Predicted Differentially Expressed Target Genes in Contrast Comparison	Predicted Differentially Expressed Target Genes Unique in EHM Horses
**Upregulated in EHM vs non-EHM horses: contrast comparison (between groups)**						
Novel (id: 764)	unknown	CCCGCCCGGCCCGGCCGCC	2.63	0.04	ND	ND
Novel	mmu-miR-7059-5p	GCCGGGGAGCCCGGCGGGC	2.01	0.03	ADAM22, CREB5, FADS2, INSR, KLF6, OAS2, ORAI2	ADAM22, ARL4D, CREB5, F11R, FADS2, HIC1, IER5, INSR, JAM2, JUND, KLF6, LAYN, ORAI2, PARD6G, PSD3, PSMB9, SOBP, SRGAP1, TBXA2R
**Downregulated in EHM vs non-EHM horses: contrast comparison (between groups)**						
Novel	mmu-miR-669k-5p	TGTGCATGTGTGCATGTAGGCAG	−1.3	0.05	ADAM22, CMPK2, CREB5, DUSP1, EPHA4, INSR, KCNC4, KCNQ4, LONRF3, MEGF9, NHSL2, NUDT4, OAS2, PAQR8, PLCB1, SCD, SLC16A14, SLC18B1, SLC7A11, STARD13, TET1	ADAM22, AGAP1, AKAP7, BAIAP3, CORO2A, CREB5, ELOVL7, EMP1, FOSB, HSPA2, HSPA5, HTRA1, INSR, INSRR, JAM2, KCNQ4, LONRF3, MEFV, MEGF9, NFASC, NHSL2, NUDT4, PLCB1, PARD6G, PLEKHA6, PSD3, PTPRF, QPCT, SDC3, SLC18B1, SLC7A11, ADAM22, CMPK2, CREB5, XCR1, DUSP1, SRGAP1, STARD13,TAP1,TOR1B, TMEM26,
eca-miR-199a/b-5p	hsa-miR-199a-3p	ACAGTAGTCTGCACATTGGTT	−1.33	0.008	ACER2, FOS, FRY, SCD SLC7A11, LONRF3, PLCB1,	ACER2, TLE2, FRY, FOS, SCD, PTPRF, CXXC5, LONRF3, PLCB1, SLC7A11, FN1
eca-miR-34c	hsa-miR-34a-5p	AGGCAGTGTAGTTAGCTGATTGC	−1.3	0.008	ADAM22, CREB5, DIXDC1, FRMD4A, KLF4, LONRF3, MAP1A	ADAM22, CERS4, CREB5, DIXDC1, DLL1, FOSB, LONRF3, PSD3, RGMB, SEMA4C, TRANK1, VAT1, VWA5B2
Novel	hsa-miR-542-5p	TCGGGGATTCAGGTGGCTGTTC	−1.24	0.008	BCORL1, GPR137C, INSR, KLF6, ORAI2, OSM, PTCH1, TEX35, ZNF862	ADAM33, ADAMTS2, FOSB, GFI1, IER5, INSR, JUND, KLF6, KLRB1, LTBP4, NECAB3, NPDC1, ORAI2, PARP12, PLEKHG5, PTCH1, RGS16, RHOB, SDC3, SRGAP1, TCF7L2, TMEM151B, TNFSF14, ZBTB16, ZFP36L2, ZNF827
eca-miR-10b	hsa-miR-10a-5p	TACCCTGTAGAACCGAATTTGT	−1.20	0.035	STARD13, EPHA4	NFASC, SOBP, STARD13, ZBTB16, ZNF827
eca-miR-328	hsa-miR-328-3p	CTGGCCCTCTCTGCCCTTCCGT	−1.13	0.01	NHSL2	CHP2, HIC1, LHFPL2, MFSD2A, NHSL2, PLEKHA6, PTPRF, TCF7L2
eca-miR-146a	hsa-miR-146a-5p	TGAGAACTGAATTCCATGGGTT	−1.05	0.001	BCORL1, SGIP1	CDS1, CNTFR, LAYN, MYBL1,SCN3B, VAT1
**Uniquely upregulated in EHM horses (within group)**						
Novel (id: 764)	unknown	CCCGCCCGGCCCGGCCGCC	2.31	0.001	ND	ND
Novel (id: 983)	unknown	CCGCCCGCCGCCGCCGCC	1.74	0.01	ND	ND
Novel (id: 187)	unknown	CCCGCCCGCCGCCGCCGCC	1.73	0.01	ND	ND
Novel	hsa-miR-7108-3p	CCCCGCCCGCCGCCGCCG	1.7	0.01	ATF3, CD7, DBP, INSR, LIPE, MAPK13, NEO1, NLRP12, ORAI2, OSCAR, OSM, PDGFB, PTCH1, SEMA6C, ZNF862	ADAR, ARL4D, AXIN2, CD180, CD7, EMP1, FOSB, GATA2, GPNMB, HIC1, HSH2D, IER3, INSR, JUND, LAMTOR2, LIPE, MAPK13, MICAL2, MSR1, NEO1, NFASC, NLRP12, NPTX1, NTNG2, ORAI2, PDGFB, PLEKHG5, PLXND1, PSMB9, PTCH1, PTPRF, SDC3, SEMA4C, SEMA6C, SOBP, TNFSF14, VAT1, ZBTB16, ZDHHC1
Novel	mmu-miR-7059-5p	GCCGGGGAGCCCGGCGGGC	1.6	0.002	ADAM22, CREB5, FADS2, INSR, KLF6, OAS2, ORAI2	ADAM22, ARL4D, CREB5, F11R, FADS2, HIC1, IER5, INSR, JAM2, JUND, KLF6, LAYN, ORAI2, PARD6G, PSD3, PSMB9, SOBP, SRGAP1, TBXA2R
**Uniquely downregulated in EHM horses (within group)**						
eca-miR-483	unknown	CACTCCTCTCCTCCCGTCTTCT	−1.8	<0.001	ND	ND
eca-miR-146a	hsa-miR-146a-5p	TGAGAACTGAATTCCATGGGTT	−1.49	<0.001	BCORL1, SGIP1	CDS1, CNTFR, LAYN, MYBL1,SCN3B, VAT1
eca-miR-34c	hsa-miR-34a-5p	AGGCAGTGTAGTTAGCTGATTGC	−1.34	<0.001	ADAM22, CREB5, DIXDC1, FRMD4A, KLF4, LONRF3, MAP1A	ADAM22, CERS4, CREB5, DIXDC1, DLL1, FOSB, LONRF3, PSD3, RGMB, SEMA4C, TRANK1, VAT1, VWA5B2
eca-miR-138	hsa-miR-138-5p	AGCTGGTGTTGTGAATCAGGCCG	−1.31	<0.001	LONRF3, DAPK2, FRMD4A, FRMPD3	AHDC1, DAPK2, EXT1, FOSB, LONRF3, NPTX1, SCN3B, SDC3, SEMA4C, SOBP, ZFP36L2
eca-miR-199b-5p	hsa-miR-199a-3p	ACAGTAGTCTGCACATTGGTT	−1.03	<0.001	ACER2, FOS, FRY, SCD LONRF3, PLCB1, SLC7A11	ACER2, TLE2, FRY, FOS, SCD, PTPRF, CXXC5, LONRF3, PLCB1, SLC7A11, FN1

Table 3. Differentially expressed (FDR < 0.05 and log 2-fold change >|1|) miRNAs are shown here. The top lists indicate miRNAs identified in the between group (contrast) comparison. Up-regulated/downregulated terms are those based on the genes upregulated/downregulated in EHM horses compared to Non-EHM horses. The bottom lists indicate the miRNAs uniquely differentially expressed within the EHM group (pre vs. post CH). Upregulated/downregulated refers to genes upregulated/downregulated during viremia compared to pre CH. There were no differentially expressed miRNAs in the non-EHM horses pre vs. post CH. ID numbers next to novel miRNAs indicate the arbitrary ID given to those without corresponding equine, murine, or human IDs. Predicted target genes that were also differentially regulated in the contrast comparison or uniquely regulated in EHM horses are indicated in red when significantly upregulated and in black when significantly downregulated. A summary of predicted target genes of differentially up- and downregulated miRNAs are shown and how they overlap with differentially expressed genes unique to EHM horses in response to EHV-1 CH (within group comparison) and between groups (contrast comparison) are shown in Figure 13.

## Data Availability

The raw data presented in this study are openly available and have been in deposited under SRA BioProject number PRJNA705083.

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
