# Peer review of "Identification of Host Factors Associated with the Development of Equine Herpesvirus Myeloencephalopathy by Transcriptomic Analysis of Peripheral Blood Mononuclear Cells from Horses"

_viruses, 2021, doi:10.3390/v13030356_

Round 1
Reviewer 1 Report
The authors conducted an extensive study to investigate factors associated with the development of equine herpesvirus myeloencephalopathy (EHM) using an "old mare" model. The scientific approach and methods are sound and the authors collected a robust dataset providing new information that adds to the body of knowledge about EHV-1 pathogenesis and EHM.
A few minor points;
1) the mansuscript would benefit if all of the tables were moved to supplementary material. Many of the tables list sequencing information that is summarized in a figure or described adequately in the text.
2) the authors measured viremia be quantitating virus genome copies. While this is useful information, a better measure of infectious particles in the blood and PBMC would be obtained from running plaque assays.
Author Response
see attached for a response to the reviewers comments

Reviewer 2 Report
The authors experimentally induced EHM among a group of horses by infecting 7 young horses and 7 older horses with EHM-associated variant of EHV-1. They produced EHM among 8 horses, the 7 older mares and one young mare. They compared the transcriptome for PBMC and compared the two groups (EHM + vs EHM-). The work is well described, well controlled and well analyzed. The authors support the observation that immunosenescence may play a role and discuss the results in terms of changes in gene expression for the two groups.
I only have a few, small editorial suggestions:
- Line 147, the authors identify MicroRNA and should introduce (miRNA) at this point. They refer to this material as miRNA in most of the rest of the article.
- Line 154. They mention the small RNA libraries and I assume they meant the miRNA libraries?
- Line 195: The authors should identify the difference in the content of Figures 1 and 2, just for clarity. Perhaps just “…. described in Figures 1 (mRNA) and 2 (miRNA).”
Author Response

(The authors gave the same response as above.)
